# CESAR: Automatic Induction of Compositional Instructions for Multi-turn Dialogs

**Taha Aksu**[1][†][*]**, Devamanyu Hazarika**[2][†]**, Shikib Mehri**[2]**, Seokhwan Kim**[2]**,**
**Dilek Hakkani-Tür**[2]**, Yang Liu**[2]**, Mahdi Namazifar**[2]
[1]National University of Singapore
[2]Amazon Alexa AI
taksu@u.nus.edu, dvhaz@amazon.com

## Abstract

Instruction-based multitasking has played a critical role in the success of large language models (LLMs) in multi-turn dialog applications. While publicly-available LLMs have shown promising performance, when exposed to complex instructions with multiple constraints, they lag against state-of-the-art models like Chat-GPT. In this work, we hypothesize that the availability of large-scale complex demonstrations is crucial in bridging this gap. Focusing on dialog applications, we propose a novel framework, CESAR, that unifies a large number of dialog tasks in the same format and allows programmatic induction of complex instructions without any manual effort.

We apply CESAR on InstructDial, a benchmark for instruction-based dialog tasks. We further enhance InstructDial with new datasets and tasks and utilize CESAR to induce complex tasks with compositional instructions. This results in a new benchmark called InstructDial++, which includes 63 datasets with 86 basic tasks and 68 composite tasks. Through rigorous experiments, we demonstrate the scalability of CESAR in providing rich instructions. Models trained on InstructDial++ can follow compositional prompts, such as prompts that ask for multiple stylistic constraints.

## 1 Introduction

Instruction tuning is a popular multi-tasking method for fine-tuning large language models (LLMs). In this setup, LLMs are trained over a range of tasks specified by instructions, which lets them generalize over new task descriptions at ease (Wei et al., 2021). Although instruction tuning enables individual task performance at scale, a language model's practical usage often requires high performance on *compositions* of these tasks. For example, in Fig. 1, the prompt requires the

---

* Work done during an internship at Amazon Alexa AI.
† First two authors contributed equally.

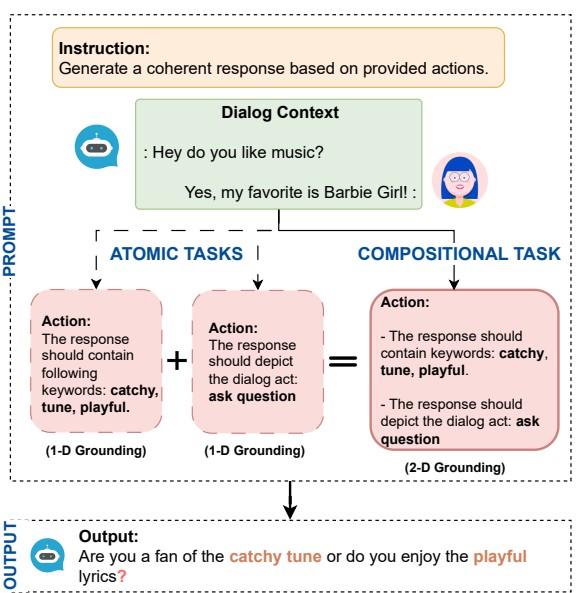

Figure 1: An illustrative integration of compound tasks, namely *keyword controlled generation* and *act grounded generation*, into a more complex compositional one. These tasks are automatically merged–without human oversight–using CESAR.

model's response to meet two control dimensions, $(i)$ incorporating three keywords in the response — *'catchy', 'tune'*, and *'playful'* and $(ii)$ following the dialog act — *ask question*. A large language model (LLM) may perform well at these dimensions individually. However, it may struggle to meet the requirements simultaneously as it has not seen such a composition of constraints during the training process. Prior work has addressed this problem through novel architectures or prompting tricks (Peng et al., 2023; Ramakrishnan et al., 2022; Hu et al., 2022). However, despite the proven effectiveness of scaling up the number of tasks (Chung et al., 2022a), the prior efforts, to the best of our knowledge, have yet to focus on scaling up compositional data during training.

One could handle complex instructions by introducing compositional tasks as demonstrations at

| Collection | Size | Prompt | # Tasks | # Data Points | Objective |
|---|---|---|---|---|---|
| FLAN | 10M-540B | Zero Few COT | 1836 | 15M | NLP Tasks |
| OPT-IML | 30-175B | Zero Few COT | 2067 | 18M | NLP Tasks Meta Tasks |
| InstructDial | 400M-3B | Zero Few | 48 | 250k | Dialog Tasks |
| CESAR | 3B-11B | Zero | 154 | 450k | Dialog Tasks Compositional Tasks |

Table 1: CESAR compared against other recent instruction tuning benchmarks. *Zero* and *Few* stand for zero- and few-shot, respectively, whereas COT stand for chain of thought prompting.

training stages. However, getting compositional data at scale is a non-trivial task because the number of compositions grows exponentially with the number of atomic tasks. This introduces significant human labor in adding appropriate instructions for each new composition.

A naive solution to this challenge might be *combining* individual task prompts' instructions and control sequences, following the controlled generation literature (Yang et al., 2022; Liu et al., 2022). However, for cross-task compositions with multiple constraints, this could result in nonsensical tasks, in ways such as their composition is either infeasible, invalid or too idiosyncratic to be of any practical use (see Fig. 10 for an example). Thus, reliable and scalable mechanisms to compose tasks (and their instructions) without manual effort are highly desirable.

**Contributions.** To address the above-mentioned challenge, we make the following contributions:

*i*) First, we propose an instruction-based framework for dialog tasks – named CESAR. CESAR modularizes dialog task prompts based on their input, constraints, and outputs. This enables an automatic combination of specific parts of different task prompts to generate compositional task prompts without any human intervention — *e.g.* in Fig. 1 CESAR combines two atomic tasks which only differ in the constraint component of the prompt. We describe the complete framework in §4.

*ii*) We introduce InstructDial++, an update over the original InstructDial benchmark (Gupta et al., 2022). It incorporates more atomic tasks and datasets and introduces composite tasks by utilizing the CESAR framework. Overall, InstructDial++ consists of 86 basic and 68 composite tasks defined on 63 datasets. We detail the InstructDial++ benchmark in §5.

*iii*) Finally, we perform comprehensive experiments that reveal multi-faceted benefits of having compositional tasks in fine-tuning stages (*c.f.* §6): (*a*) They improve compositional task performance for both seen and unseen task compositions; (*b*) They improve atomic or non-compositional task performance under similar data budgets.

The CESAR framework, along with the Instruct-Dial++ benchmark, enables the automated generation of complex tasks in dialog benchmarks, which we believe is one of the critical ingredients in bridging the gap between publicly-available dialog models compared to proprietary AI assistants.

## 2 Related Work

The term *instruction-tuning* was popularized by Wei et al. (2021); Mishra et al. (2022) and has gained popularity among various researchers—for example, Ouyang et al. (2022) optimized outputs based on user preferences by incorporating human feedback. Chung et al. (2022a), on the other hand, demonstrated that scaling tasks and model size improved performance across benchmarks. Finally, InstructDial (Gupta et al., 2022) focused on instruction tuning for downstream dialog tasks but lacked curated compositional tasks. To address this gap, CESAR enables large-scale training of compositional tasks. Please refer to Table 1 for a comparison of CESAR with the other recent benchmarks[1].

**Unified Grounding.** The research community has recently emphasized Unified Grounding as a solution for diverse datasets. UnifiedSKG (Xie et al., 2022) unifies 21 SKG tasks into a text-to-text format, demonstrating improved performance through multitask learning. Convlab3 (Zhu et al., 2022) proposes a shared format for task-oriented

---

[1]While Flan and OPT also consider dialog datasets, its coverage is not as extensive as InstructDial and CESAR.

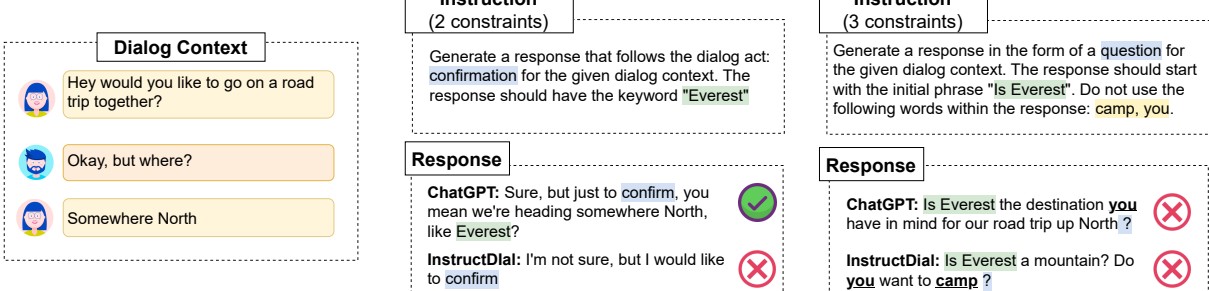

Figure 2: Investigating the disparity between instruction following capability for dialog response generation between closed-access (ChatGPT) and open-access (InstructDial) dialog models. ChatGPT comfortably satisfies two constraints in dialog response generation whereas showing signs of struggle in $>= 3$ constraint scenarios (more examples in Appendix B). In contrast, InstructDial lags behind ChatGPT since it is unable to satisfy $>= 2$ constraint scenarios (further evidence in Fig. 3).

dialog datasets. BlenderBot 3 (Shuster et al., 2022) adopts a modular approach, assigning specific tasks to different parts of a prompt. However, none of these frameworks have explored task composition from their unification efforts.

**Compositional Generalization via Structure.** Incorporating structure into the prompts have consistently been shown to enhance the language model's ability to handle compositional tasks. Bursztyn et al. (2022) propose compositional fine-tuning, which involves breaking down the task into its constituent parts and combining these parts using decision templates. Keysers et al. (2019) use logical forms in training data to enable rule-based generation of compound samples for compositional generalization in semantic parsing tasks. Finally, Chen et al. (2022b) design task-specific prompts that provide detailed information about each task to help the model understand commonalities across different tasks.

**Dialog Control.** Various approaches have been used to control multiple attributes, such as adhesive architectures or prompting techniques. For instance, Chen et al. (2022a) define task-specific prompts and concatenate them at runtime to tackle complex tasks. Alternatively, SHAO et al. (2023) learn a compositional codebook where each task corresponds to a combination of codes, allowing controllability at inference time. Subramanian et al. (2018) embed each target attribute separately and average them as the start-of-sequence symbol during generation. Hu et al. (2022) propose a two-stage decoder that imposes stylistic and word-level constraints separately within the seq2seq framework. However, none of these approaches directly

compare to CESAR framework, which specifically addresses the scalability of compositional tasks in training data.

## 3 Motivation

In this section, we first explore the performance disparity between closed-access and open-access models [2] on complex compositional tasks. We then investigate the impact of including compositions in the training data on task performance.

**Closed- vs. Open-access Models in Complex Dialog Tasks.** We begin by checking the disparity between open and closed-access models. In Fig. 2, we notice that ChatGPT (a closed-access model) can produce satisfactory results for simple composite tasks whereas publicly available DIAL-T0 (Gupta et al., 2022) struggles. This demonstrates that open-access models require additional resources to improve on complex dialog tasks.

**Can Compositional Demonstrations Improve Performance?** Next, we want to verify whether the presence of compositional demonstrations can improve performance on complex dialog tasks. For this, we design a preliminary experiment where we select four dialog tasks: generating dialog responses where we control the $i)$ beginning phrase, $ii)$ the ending phrase, $iii)$ the length (short, medium, and long), and $iv)$ keyword to be incorporated in the response. We manually create instructions for all possible combinations of these tasks (*e.g.* combining *begins with* with *ends with* generation, amongst others. We fine-tune the public

---

[2]We define closed vs. open-access based on whether the model parameters and data are publicly available or not.

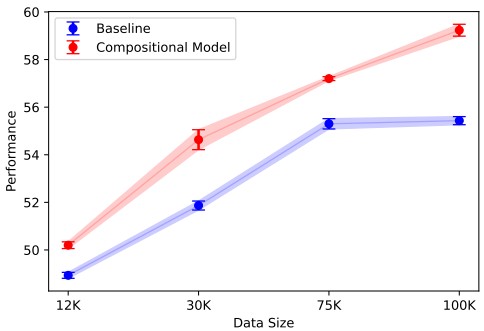

Figure 3: Compositional accuracy of both the baseline and compositional model over a varying number of training data sizes. Each datapoint is run across three independently sampled test sets to account for variability — *c.f.* Fig. 8 for atomic performance comparison.

| | Dialog Components | Sample Dialog Items |
|---|---|---|
| **C** | Dialog context between the two speakers. | utterances |
| **S** | State of the dialog context. | dialog summary, speaker intent, etc. |
| **E** | Evidences that could be relevant for the response. | retrieved knowledge, persona, etc. |
| **A** | Actions/constraints that the response has to follow. | utterance style, dialog act, etc. |
| **R** | The next response by the assistant. | response utterance |

Table 2: Dialog Components of CESAR with sample Dialog Items.

Flan-T5-xl model (Chung et al., 2022b) on two different sets of training data, each with the same size, to create two models: $i)$ `Baseline` – trained solely on the four atomic tasks, and $ii)$ `Compositional` – trained on a mixture of atomic and compositional tasks. For a fair comparison, we keep the training steps the same for both models.

In Fig. 3, we observe that `Compositional` model outperforms `Baseline` model, regardless of the training size. This indicates that including compositional tasks in the training data is crucial for better performance. Interestingly, we also find that the presence of compositional tasks in the training data positively affects atomic task performance ( Fig. 8). This analysis reveals the need for a scalable generation of compositional dialog tasks and instructions – a gap that we fill with CESAR.

## 4 CESAR

For any given dialog interaction, we first define the notions of *dialog items* and *dialog components*:

- **Dialog items** are units of information pertain-

ing to a dialog interaction, such as *utterances*, *speaker states*, *intents*, *personas*, *stylistic attributes* of utterances, *dialog summaries*, *utterance revisions*, *external knowledge snippets*, amongst others.

- **Dialog Components** are logical categories of a dialog, which include: $C \rightarrow$ *context* (or dialog context); $S \rightarrow$ dialog state(s); $E \rightarrow$ evidence(s); $A \rightarrow$ action(s); $R \rightarrow$ the dialog response. Any dialog item $\lambda$, can be mapped to a dialog component using a mapping function $g()$, i.e., $g(\lambda) \rightarrow \{C, S, E, A, R\}$. Table 2 provides an overview of the dialog components along with some sample dialog items that are mapped to it.

### 4.1 CESAR Framework

CESAR adopts a structured approach to dialog processing, often seen in task-oriented dialogs (Hosseini-Asl et al., 2020). It generalizes the rigid structure of task-oriented dialogs to accommodate natural dialogs. A task in the CESAR framework is essentially a text-to-text task where both the input and the output are a *linearized* [3] representation of multiple dialog components formatted as per a specified structure. A high-level representation of a CESAR task is defined as follows:

$$IC\Lambda - \psi$$
$$= IC\underbrace{\{g(\lambda_1), \dots, g(\lambda_m)\}}_{\text{grounding}} - \psi, \qquad (1)$$

where, the symbol '$-$' delineates the input from the output, i.e. {input_prompt} $-$ {output}.

**Input:** The input prompt in a CESAR task contains three prime components. First, $I$ represents a task instruction. Second, $C$ includes the dialog context, which are previous utterances by the two speakers. Finally, $\Lambda = \{g(\lambda_1), \dots, g(\lambda_m)\}$ is the multiset of dialog components[4] that is provided in the input in addition to the instruction $I$ and the dialog context $C$.

**Output:** $\psi \in \{S, E, A, R\}$ represents the task output described by the instruction $I$ and context $C$. Note that $\psi$ is never empty, as each task needs an output. [5]

---

[3]We aim to achieve order invariance in this linearization, using a random sampling process described in §5.2.

[4]A multiset is a set allowing multiple instances of its items.

[5]We limit the scope of this work to tasks with only one component as output. Naturally, there could be tasks (or in-

|  | Speaker | Utt. | State | Evidence | Action |
|---|---|---|---|---|---|
| Input | User | $r_1$ | $\{s_{11}, s_{12}\}$ | $\{\}$ | $\{a_{11}\}$ |
|  | Assistant | $r_2$ | $\{s_{21}\}$ | $\{e_{21}\}$ | $\{a_{21}\}$ |
|  | User | $r_3$ | $\{s_{31}, s_{32}\}$ | $\{\}$ | $\{a_{31}\}$ |
| Output | Assistant | $r_4$ | $\{s_{41}\}$ | $\{e_{41}, e_{42}\}$ | $\{a_{41}, a_{42}\}$ |

(a) For any dialog item $x_{ij}$, $i$ refers to it's turn number in the dialog and $j$ refers to it's identification within the same dialog component, i.e., S, E, A, or R. for that turn. Fig. 4 provides an example for this setup.

| CESAR Task | Input | | Output |
|---|---|---|---|
|  | C | $i$-D, $\Lambda$ | $\psi$ |
| IC-S | $\{r_1, r_2\}$ | 0-D, $\{\}$ | $s_{21}$ |
| IC-A | $\{r_1, r_2\}$ | 0-D, $\{\}$ | $a_{31}$ |
| IC-E | $\{r_1, r_2, r_3\}$ | 0-D, $\{\}$ | $e_{41}$ |
| ICS-A | $\{r_1, r_2\}$ | 1-D, $\{s_{21}\}$ | $a_{31}$ |
| ICE-A | $\{r_1, r_2, r_3\}$ | 1-D, $\{e_{41}\}$ | $a_{41}$ |
| ICA-R | $\{r_1, r_2, r_3\}$ | 1-D, $\{a_{41}\}$ | $r_4$ |
| ICEA-R or ICAE-R | $\{r_1, r_2, r_3\}$ | 2-D, $\{e_{41}, a_{41}\}$ | $r_4$ |
| ICSE-R or ICSE-R | $\{r_1, r_2, r_3\}$ | 2-D, $\{s_{31}, e_{41}\}$ | $r_4$ |

(b) For the given input dialog context $\{r_1, r_2, r_3\}$ and output dialog response $\{r_4\}$ in Table 3a, we provide some example tasks defined under CESAR framework.

Table 3

Let us provide a concrete example of the CESAR framework. Imagine that a *user* is interacting with an AI *assistant*, which is shown in Table 3a. This dialog includes different *dialog items*. For example, evidence item $e_{41}$ that is useful to construct the utterance $r_4$ or state item $s_{32}$ that either infers information about utterance $r_3$ or the complete dialog history $r_1, r_2, r_3$. Given these dialog items, we can construct several CESAR tasks, some of which are demonstrated in Table 3b. We also illustrate an example dialog interaction, including dialog items with their dialog component mapping in Fig. 4.

## 4.2 CESAR Tasks

We now define an n-D CESAR task:

> **Definition 1** (*n-D Task*)**:** For any CESAR task of the form $IC\Lambda - \psi$, we call the task n-D Task if there are $n$ dialog items in $\Lambda$, i.e. $|\Lambda| = n$.

Table 3b illustrates multiple CESAR tasks that are 0-D, 1-D, and 2-D tasks framed from the dialog items in Table 3a. Note that a 0-D task does not mean the input is empty, since a CESAR task always assumes a task instruction $I$ and a dialog context $C$ (which can be potentially empty if there is no previous interaction).

**Atomic vs. Compositional Task.** In CESAR, we categorize every task as either *atomic* and *compo-*

structions) where a user could ask to generate multiple outputs, such as generating a dialog summary along with an appropriate response. We defer such a setup for future work.

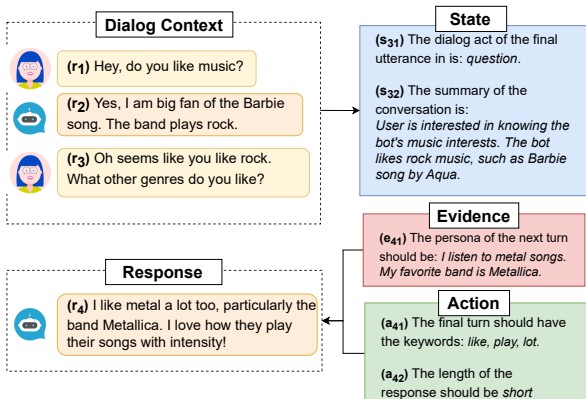

Figure 4: An example dialog with dialog items: utterances $(r_1, r_2, r_3, r_4)$, state $(s_{31}, s_{32})$, evidence $(e_{41})$, actions $(a_{41}, a_{42})$ as described in Table 3a.

*sitional* task: Atomic Tasks are either 0-D or 1-D tasks; Compositional Tasks are any n-D Task with $n \geq 2$.

To create any compositional task in CESAR, we define the following composition operation.

> **Definition 2** (*Task Composition*)**:** *For two $i$-D Tasks,*
> $$IC\left(\Lambda \cup \{g(\lambda_a)\}\right) - \psi, \text{ and}$$
> $$IC\left(\Lambda \cup \{g(\lambda_b)\}\right) - \psi,$$
> *where, $|\Lambda| = i - 1$ and $i \geq 1$, we combine the two tasks to form an $(i + 1)$-D Task:*
> $$IC\left(\Lambda \cup \{\lambda_a, \lambda_b\}\right) - \psi$$

This composition operation allows the creation of arbitrarily complex compositional tasks, i.e., m-D Tasks with $m \geq 2$, subjected to the availability of relevant atomic tasks.

Our proposed CESAR framework can also incorporate dialog items as reasoning elements in chain of thought (Wei et al., 2022). We present this extended formulation in Appendix E, but keep its experimentation as a future work.

## 5 InstructDial++

InstructDial is an instruction tuning benchmark for dialogue, which consists of a repository of diverse dialogue tasks in a unified text-to-text format created from openly available dialogue datasets (Gupta et al., 2022). In this section, we first discuss new tasks and datasets added to this benchmark, resulting in the updated version, InstructDial++. We then discuss how we map each task in InstructDial++ to the CESAR framework.

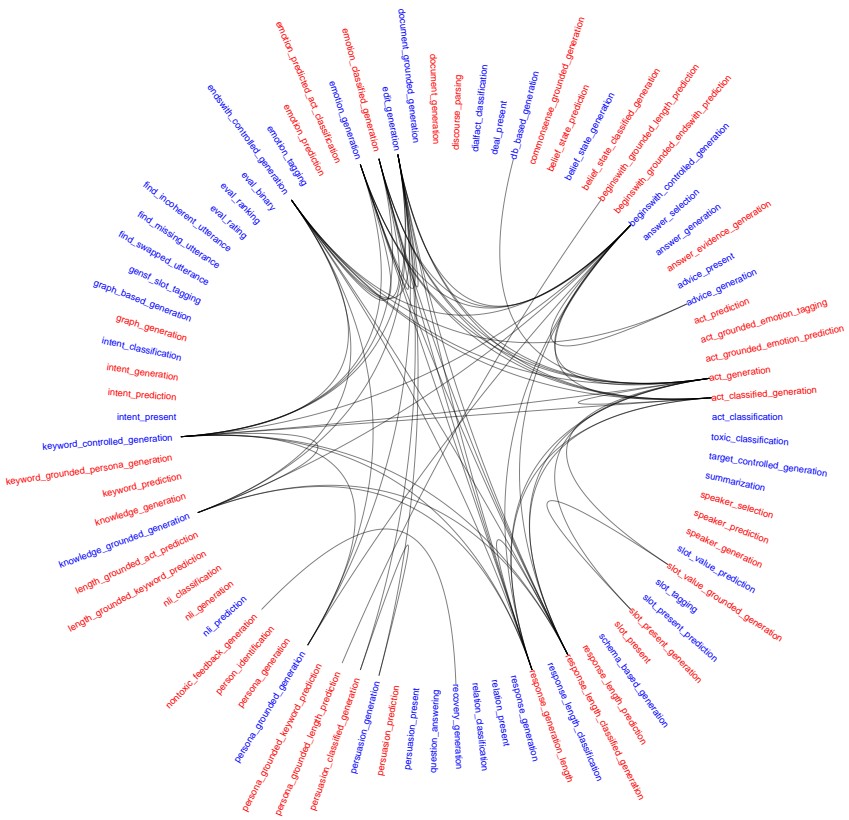

Figure 5: Chord Wheel showing all atomic and compositional tasks in CESAR. Tasks colored in red are newly added compared to the InstructDial dataset. Each edge between a pair of tasks indicates a new compositional task that combines them together.

## 5.1 New Tasks and Datasets

Knowing that scaling data in training benchmarks has a positive impact on performance (Longpre et al., 2023), we expand the InstructDial benchmark by incorporating 15 additional datasets and 42 new atomic (*i.e.* 0-D & 1-D) tasks (*c.f.* Table 15).

The majority of newly introduced atomic tasks are derived from the inherent structure of the CE-SAR framework, which allows for multiple tasks to be defined and performed on the same dataset. For example, we create 8 novel tasks (not comprehensive) listed in Table 3b using the dialog depicted in Fig. 4. Following the inclusion of the newly added tasks, the Instructdial++ benchmark consists of 86 tasks across 65 datasets — *c.f.* Fig. 5 and Table 15. The next step is to explain how we have mapped each Instructdial task to the CESAR format.

## 5.2 Mapping InstructDial++ to CESAR

To map each InstructDial task into the CESAR format, we begin by assigning them a CESAR task based on the input constraints and output type. None of the tasks in Instructdial++ incorporate reasoning or CoT, leading to the simplified CESAR

format $IC\Lambda - \psi$ (Eq. (1)).

**Prompt Design.** Unlike InstructDial's approach of providing unique instructions for each task, we adopt a more general approach in crafting our instructions which cover two main aspects: $i$) identifying the dialog components (*c.f.* Table 2) to focus on during generation, and $ii$) specifying which component the generation is aimed at, such as the example instruction: *Provide the correct value for action field given the dialog context, state, and evidence fields.* Due to the structural form of the instruction, we can programmatically generate instructions for each compositional task.

**Generative and Discriminative Tasks.** Despite the generative nature of each CESAR task, it is important to note that our framework enables us to specify discriminative tasks as well. As an illustration, in the *emotion_classification* task, we provide the candidate emotions by incorporating them within the state component, *e.g.* 'State: Candidate emotions are sad, happy, and mad. The emotion of the last utterance is:'.

CESAR framework enables 4 0-D tasks, and 12

1-D tasks. InstructDial++ incorporates downstream tasks for each 0-D grounding task and for 10 of all the 1-D grounding tasks as depicted in Fig. 6. Please find examples of the input-output structure of these tasks in Table 3b

**0-D Tasks.** The top of Fig. 6 depicts all 4 0-D tasks. Each of these CESAR tasks clusters downstream tasks with a similar output objective. For instance, IC-S tasks involve categorizing, formatting, or organizing information within the dialog context in a more succinct or structured way, *e.g. dialog-summarization*. IC-E, on the other hand, involves generating external knowledge useful in the dialog's context; both *persona generation* and *knowledge generation* are downstream tasks under this category. We believe the persona information of a user is better fed as 'evidence' rather than an 'action' because it is external information and not a strict constraint to follow during generation, *c.f* Table 2. IC-A tasks are responsible for generating actions to be followed in the response, such as *keyword* or *intent* prediction. Finally, IC-R is the collection of tasks that generate/select a response for a given dialog context.

**1-D Tasks.** 1-D tasks have the same categorization because their generation is also aimed at one of S, E, A, R components as in 0-D tasks, except they ground on an additional component other than the dialog context. For example, for ICA-R, the response generation task is additionally conditioned on a provided action, *e.g. begins-with controlled generation* or *slot-value grounded generation*. We also include *edit generation* under this category because, unlike an IC-R task, it grounds on both the context and the previous version of the response to be corrected. Another 1-D CESAR task example is ICS-A which involves generating action of the upcoming response conditioned on the state of the current dialog context. An illustrative example is *length-grounded keyword prediction*, where the generated keywords (for the response) are conditioned on the length of the final utterance in the dialog context. As depicted in Fig. 6, InstructDial++ incorporates 7 more 1-D tasks along with the 2 that we borrow from InstructDial.

**2-D Tasks.** "After manually mapping all tasks of 0-D and 1-D nature, the CESAR framework selects and organizes viable 2-D tasks automatically according to a small subset of predefined rules. These rules ensure the combi-

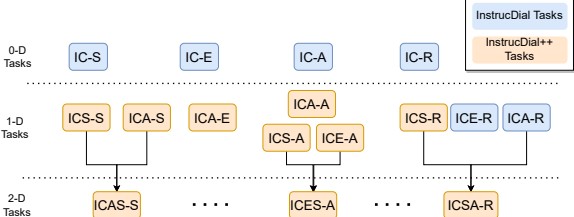

Figure 6: Comparsion of CESAR tasks in InstructDial++ vs. their counterparts in InstructDial.

nation does not result in an infeasible task — *c.f.* Fig. 10. One sample rule allows compositions where the generation incorporates 2 actions (e.g. beginswith_controlled_generation and keyword_controlled_generation) creating ICAA-R task. For a comprehensive explanation of these rules please see Appendix C

This results in 68 new downstream tasks defined on 7 2-D Cesar tasks — *c.f. Table 16 and Fig. 11*. Fig. 5 shows each of these compositions as edges between atomic tasks.

**Order Invariance of Grounding Items** As dialog items in a prompt must be linearized, we adopt a randomizing process to ensure that our models are invariant in ordering the items. We place each section randomly inside the prompt, with two specific rules: $i$) the instruction is always placed at the beginning, which is a common practice; $ii$) the target section, referred to as $\psi$ in Eq. (1), is placed at the end. The rest of the sections (C and $\Lambda$ from Eq. (1)) are randomly positioned within the prompt.

# 6 Experiments

## 6.1 Setup

**Models.** Throughout experiments, we utilize ChatGPT model gpt-3.5-turbo-16k-0613 and five public models, namely $i$) T0-3B (Sanh et al., 2021) which is trained on a mixture of downstream tasks, $ii$) DIAL-T0 and $iii$) DIAL-BART0 (Gupta et al., 2022), fine-tuned on InstructDial dataset and based on T0-3B and BART0 (Lin et al., 2022), respectively. We train another baseline model based on InstructDial dataset using FLAN-xxl (Chung et al., 2022a) and name it with the same convention as the authors as $iv$) DIAL-FLAN-xxl. Our main model, $v$) CESAR-FLAN-xxl, is also trained using the FLAN-xxl model but on the Instructdial++ dataset rich in compositional tasks — *c.f.* Appendix A for training details.

## 6.2 Tasks and Metrics

**Atomic Tasks.** Throughout experiments, we test models on eight atomic tasks — either individually or as part of some composition: *Begins With Generation* (**BW**): Generate a response that starts with a given phrase. *Ends With Generation* (**EW**): Generate a response that ends with a given phrase. *Keyword Controlled Generation* (**KC**): Generate a response which incorporates given set of keywords. *Length Controlled Generation* (**LC**): Generate a response with a certain length (short/medium/long). *Persona Based Generation* (**PB**): Generate a response based on a given speaker persona. *Knowledge-Based Generation* (**KB**): Generate a response based on some external knowledge. *Edit Generation* (**EG**): Edit a response to make it coherent with the dialog context. *Emotion-Grounded Generation* (**EMG**): Generate a response that depicts a certain emotion. To maintain standardized evaluation, we utilize the atomic task metrics implemented by Gupta et al. (2022) for all of our atomic tasks.

**Compositional Task Metrics.** We evaluate compositional task performance on nine tasks, which are binary compositions of the atomic tasks. In InstructDial++, these compositions are available in the test set. For InstructDial, we manually create instructions for each task composition following the same formatting as the InstructDial paper and generate new test sets accordingly.

For each compositional task performance, we report accuracy scores if possible. If the combined accuracy is unclear, we provide multiple metrics evaluating each dimension separately. For example, for *persona-based + ends with generation*, because it is difficult to quantify how well the persona was used in the final generation, we report both the "ends with accuracy" and Rouge-L metrics.

## 6.3 Results

In this section, we present the results of three main experiments. Each experiment's test set prompts for each model are formatted according to their training data. For T0-3B and DIAL-FLAN-xxl models, the prompts included natural phrases that explain the task. For DIAl-BART0 and DIAL-T0, we use the special tokens defined by Gupta et al. (2022), and for CESAR-FLAN-xxl, the prompts are automatically generated in CESAR format by the framework itself. Despite their format, each test set is composed of the same data instances from the test splits of the corresponding datasets.

**Atomic Task Performance.** Table 4 presents the atomic task performance of each model. Even though we do not claim any discrete advantage in atomic task performance, we observe that the atomic task performance of the CESAR model proves to be comparable and even better in many tasks compared to the baselines. This aligns with preliminary experiments insights, *c.f.* Fig. 8.

**Compositional Task Performance.** Table 5 presents compositional task experiments results. Our model outperforms the baselines on every task composition. This indicates that good performance on atomic tasks does not necessarily translate to good performance on compositions of those tasks, as evidenced by the widening gap between CESAR and baselines from atomic to compositional tasks. We add two qualitative examples depicting atomic and compositional generation by CESAR, interested readers can find them in Fig. 12.

**Generalization Experiment.** To evaluate the robustness capabilities of CESAR, we design a simple training setup where each model trains only on a limited set of tasks using the smaller `FLAN-xl` model. We then evaluate these models with various task compositions, both seen and unseen.

We employ three models for this experiment. $i$) `Atomic Model`, solely trained on four atomic tasks: BW, KC, LC, and EG, representing the lower bound without any compositional training, $ii$) `Naive Composer`, trained on the same four atomic tasks as well as three compositional tasks: BW+KC, KC+LC, and BW+LC. These compositional tasks are created by concatenating instructions and constraints from individual tasks within the same prompt using the conjunction 'and'. To avoid generating infeasible tasks due to the lack of structural information inherent in the CESAR framework, we manually select the tasks that the `Naive Composer` combines (as explained in Appendix G). Finally, we train another model using the $iii$) `CESAR` format, incorporating the same four atomic tasks and three compositional tasks as used for the `Naive Composer`.

The results, presented in Table 6, demonstrate that the CESAR structure outperforms the Naive Composer in all seen compositions and most unseen tasks and compositions. For the unseen task composition[6] at the right-most column, we incor-

---

[6]In order to check if these tasks are truly unseen we ex-

| Model | Training Set | BW | EW | KC | LC | PB | | KB | |
|---|---|---|---|---|---|---|---|---|---|
| | | *Acc* | *Acc* | *Acc* | *Acc* | *Bleu-2* | *R-L* | *Bleu-2* | *R-L* |
| T0-3B | Zero-shot | 13.12 | 15.74 | 20.14 | 43.38 | 1.44 | 7.7 | 3.34 | 10.44 |
| DIAL-BART0 | InstructDial | 84.02 | 61.46 | 87.24 | 44.92 | 4.73 | 14.5 | **10.83** | **21.9** |
| DIAL-T0 | InstructDial | **86.46** | 60.84 | 74.38 | 50.56 | 4.43 | 14 | 10 | 20.3 |
| DIAL-FLAN-xxl | InstructDial | 81.4 | 62.12 | 86.04 | 53.26 | 4.23 | 13.66 | 9.32 | 18.96 |
| CESAR-FLAN-xxl | InstructDial++ | 84.60 | **65.66** | **88.08** | **82.98** | **4.81** | **14.5** | 9.76 | 20.06 |

Table 4: Evaluation results on atomic tasks. *R-L* stands for *Rouge-L* metric, best results for each column are **bold**.

| Model | Training Set | BW + EW | BW + KC | BW + LC | EW + KC | EW + LC | KC + LC | PB + EW | | EG + EW | |
|---|---|---|---|---|---|---|---|---|---|---|---|
| | | Acc | Acc | Acc | Acc | Acc | Acc | EW Acc | R-L | EW Acc | R-L |
| T0-3B | Zero-shot | 2.38 | 4.39 | 5.75 | 2.95 | 6.92 | 8.55 | 5.4 | 11.56 | 22.5 | 29.9 |
| DIAL-BART0 | InstructDial | 69.01 | 77.2 | 38 | 46.3 | 11.8 | 16.65 | 82.8 | 50.2 | 85.2 | 83.2 |
| DIAL-T0 | InstructDial | 72.9 | 72.8 | 42.5 | 49.1 | 27.4 | 35.2 | 76.9 | 46.1 | 84.5 | 77.2 |
| DIAL-FLAN-xxl | InstructDial | 70.1 | 78.53 | 44.25 | 58.04 | 30.15 | 44.4 | 82.53 | 49.05 | 89.3 | 85.33 |
| CESAR-FLAN-xxl | InstructDial++ | **75.18** | **83.08** | **70.43** | **62.7** | **47.2** | **68.6** | **88.1** | **51.9** | **94.8** | **93.10** |

Table 5: Evaluation results on compositional tasks. *R-L* stands for *Rouge-L* metric, best results for each column are **bold**.

| Model | Seen Compositions | | | Unseen Compositions | | | | | Unseen Tasks | | |
|---|---|---|---|---|---|---|---|---|---|---|---|
| | BW+ KC | KC+ LC | BW+ LC | EW+ EG | | EW+ KC | BW+ EW | EW+ LC | EW+ EMG | | |
| | Acc | Acc | Acc | KC Acc | R-L | Acc | Acc | Acc | EW Acc | R-L | ChatGPT E.C. \| R.Q. |
| Atomic Model (FLAN-xl) | 73.88 | 43.8 | 50.3 | 82.2 | 69.2 | **36.1** | 54.21 | 20.1 | 49.3 | 22.6 | 60.9    1.76 |
| Naive Composer (FLAN-xl) | 74.08 | 44.55 | 52.07 | 82.5 | 73.69 | 35.24 | 54.21 | 20.35 | 50.9 | **24.2** | 63.0    1.91 |
| CESAR (FLAN-xl) | **76.07** | **44.8** | **54.4** | **83.1** | **83.7** | 31.89 | **56.91** | **21.75** | **54.6** | 22.3 | 56.4    **1.99** |

Table 6: Evaluation results on seen/unseen compositions, and unseen tasks. *R-L* stands for *Rouge-L* metric, *E.C.* stands for *emotion classification*, and *R.Q.* stands for *response quality*. Best results for each column are **bold**.

porate chatGPT to classify and evaluate the generated response's emotion and the response's quality — *c.f.* Appendix F for details on the evaluation prompts used.

It is important to note that the manual selection of tasks to be composed by the *Naive Composer* overlooks an essential contribution of our framework: the ability to detect viable compositions. Therefore, this experiment only demonstrates how the robustness is affected by the CESAR structure and is not a direct comparison between the Naive Composer and our approach.

## 7 Conclusion

We propose CESAR to fill the compositional capability gap between public models and irreproducible state-of-the-art, ChatGPT. CESAR modularizes downstream dialog tasks under one format

allowing the programmatically-induced generation of complex composite instructions. Moreover, we create a new benchmark building on top of Instruct-Dial, adding new tasks and datasets and utilizing CESAR to populate composite tasks. The new, InstructDial++ includes 63 datasets with 86 atomic and 68 composite task definitions. Our framework's compositional and atomic task abilities are demonstrated in extensive experiments. These experiments reveal that our framework significantly improves the model's ability to handle complex prompts compared to previous approaches. Notably, we discover that including composite data in the training set enhances compositional performance and improves atomic performance. Additionally, we conduct a robustness experiment and find that the CESAR structure outperforms the baselines in majority of compositions as well as in unseen tasks and compositions.

## 8 Limitations

Given its large scope, our work has yet to be able to delve into many promising avenues. We dedicate this section to discussing these to benefit future

amined the FLAN collection and saw that out of 23 datasets it incorporates we only use DailyDialog within the test set. Moreover we saw that there is minimal intersection between the version of DailyDialog used by FLAN (NIv2 corpus) and the original version we used. Thus we conclude there is only minimal chance of contamination, where in the worst case 100 test instance are seen by the model.

research:

**1) Datasets in InstructDial++ are not comprehensive.** Even though we tried to increase the dataset and task scale to our best ability there are certainly more datasets that InstructDial++ would benefit from such as Contrack (Ruckert et al., 2022), PRESTO (Goel et al., 2023), *etc.*

**2) Multi-tasking with non-dialog data is not done.** Due to the large scope of our work, we limited the scope of datasets we focused on to dialog-specific ones. Previous work has shown that adding non-dialog data might help (Zeng and Nie, 2021).

**3) We have not explored negative conditions.** A true composition should incorporate all logical compositions. The challenging part of the negative condition is its evaluation.

**4) We have only experimented using the existing grounding features available in the datasets.** This limits the kinds of controls that could be done.

**5) Automatic metrics are not necessarily robust.** We try to mitigate this by choosing control signals that can be automatically measured.

**6) Our action fields are mostly lexical, and some are semantic.** A comprehensive set of actions would be better.

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

## A   Training Details

In line with Gupta et al. (2022), we create the training data by sampling 5000 instances per atomic task from both the InstructDial and Instructdial++ datasets for their respective trainings. For Instructdial++, we additionally sample 1000 instances per compositional task for each dataset generated by the CESAR framework. Input sequences are set to a maximum length of 1024 tokens, and output sequences are set to a maximum length of 128 tokens. Longer sequences are truncated to these lengths, and shorter sequences are padded. We trained both the DIAL-FLAN-xxl and CESAR-FLAN-xxl models on eight A100 GPUs, with a batch size of 10 per device and gradient accumulation steps set to 4. Both models are trained for two epochs, with a learning rate of 5e-05, and using the AdamW optimizer (Loshchilov and Hutter, 2017).

## B   Compositional Capabilities of Closed Source Models

Fig. 7 shows 3 more examples depicting ChatGPT(gpt-3.5-turbo-0301), GPT-4 and Claude-v2's compositional task capabilities. It's evident that each of these models demonstrates a certain degree of compositional capabilities, showcasing their aptitude in complex language understanding and generation. However, there are certain scenarios where each exhibits inaccuracies or nuances that deviate from expected outputs. Amongst the three, GPT-4 consistently delivers the most reliable results whereas Claude v2 exhibits comparable performance, although it occasionally makes minor errors. GPT-3.5, on the other hand, tends to fall slightly behind its successors.

## C   Composition Rules

As explained in section §4, CESAR does not only unify dialog tasks in a certain prompt structure but it utilizes this structure to combine these dialog tasks as compositional tasks automatically. This automation is achieved by defining specific rules, 10 as of the current version, that constrain which CESAR tasks can be composed together. The comprehensive list of these rules can be found in Table 7.

The first rule delineated in the Table provides an illustrative example of how the CESAR compositional tasks are defined. This particular rule is formulated to create ICAA-R compositions, implying that it combines two distinct ICA-R tasks,

generating an output that encapsulates dual actions. For instance, the combined tasks could involve *beginswith_controlled_generation* and *keyword_controlled_generation*.

In the rule structure, the key "common fields" represent modules that are recurrent across each task involved in the composition. For this rule, the common modules include the "dialog context" and "response". These shared elements are essential as they get assimilated into the final compositional task. Furthermore, the "target field" is pivotal as it indicates the expected output from both tasks involved in the composition. Hence, in this rule, the "response" field, which is anticipated from both ICA-R tasks, becomes the output of the culminating ICAA-R task. It's also important to note that due to the consistent structure prevalent across all ICA-R tasks, the framework ensures a seamless composition. This means that irrespective of the variant of ICA-R task combined, as long as the rule's constraints are satisfied, the resulting composition should be impeccable.

## D   Benhmarking ChatGPT on Atomic and Compositional Tasks

In this section we provide additional results by ChatGPT model complementing out main results tables. We have chosen not to include these results in the main paper for two reasons: (1) ChatGPT is a closed-access model, limiting scientific insights, and (2) Replicating ChatGPT results is not guaranteed due to inaccessible architecture, parameters, and the API's lack of consistency across different runs.

To query ChatGPT, we utilize the same naturally formatted prompts as those used for the DIAL-FLAN model. An example of a zero-shot prompt can be found in Table 11. The one-shot experiments follow a similar procedure but include an additional in-context example, maintaining the same prompt structure.

Table 8 and Table 9 present additional results for the ChatGPT model in the atomic and compositional task evaluations, respectively. Results depict that ChatGPT actually has relatively lower performance compared to CESAR and DIAL-FLAN-xxl and CESAR-FLAN-xxl models. However, upon analyses of the results we saw that these results maybe somewhat misleading. Because both DIAL-FLAN-xxl and CESAR-FLAN-xxl are trained on the training splits of the tested data they learned

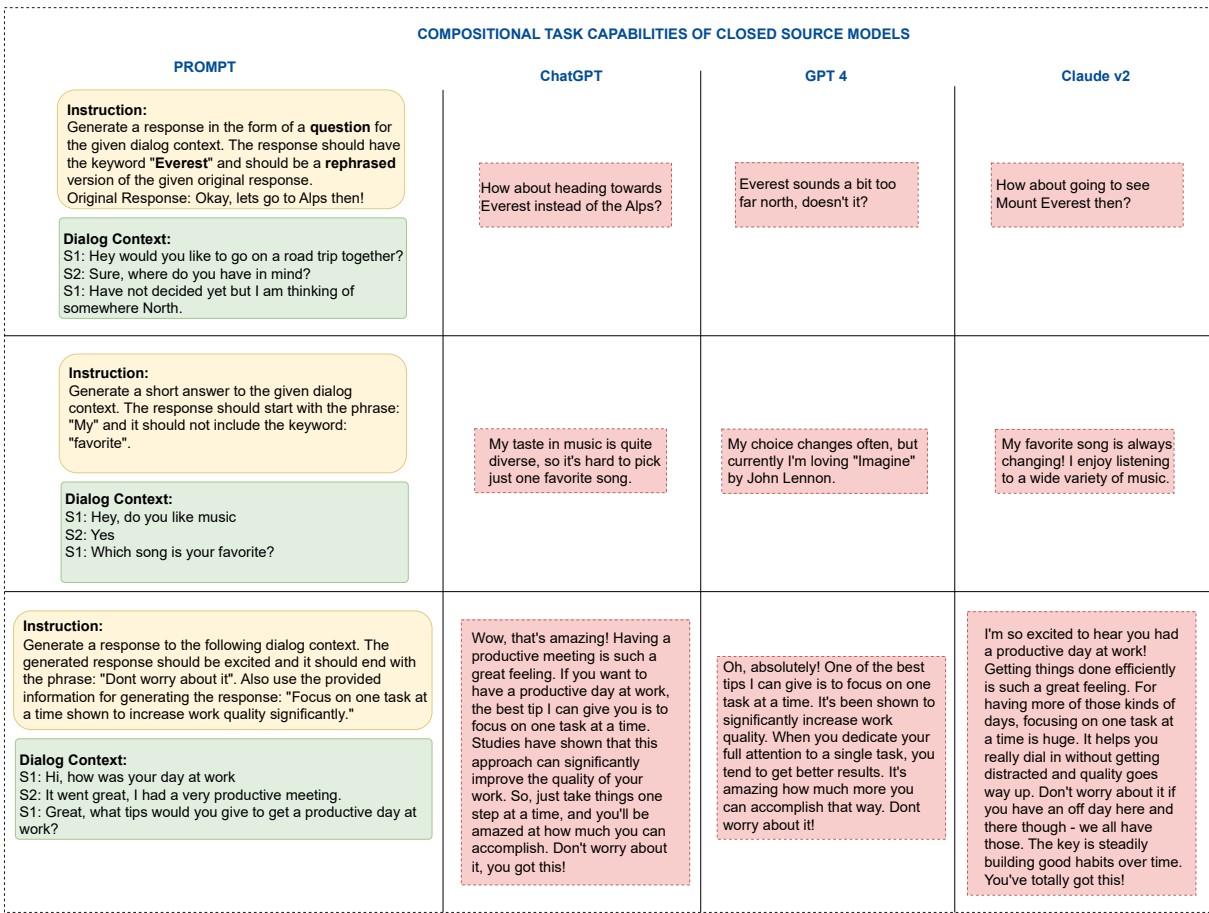

Figure 7: Qualitative examples showing compositional capabilities of some closed source models.

| Rule | Task 1 | Task 2 | Composed Task | Common Dialog Components | Target Field |
|------|--------|--------|---------------|--------------------------|--------------|
| 1 | ICA-R | ICA-R | ICAA-R | dc, r | r |
| 2 | ICE-R | ICE-R | ICEE-R | dc, r | r |
| 3 | ICE-R | ICA-R | ICEA-R | dc, r | r |
| 4 | ICS-R | ICE-R | ICSE-R | dc, r | r |
| 5 | ICS-R | ICA-R | ICSA-R | dc, r | r |
| 6 | ICS-R | ICS-R | ICSS-R | dc, r | r |
| 7 | ICE-A | ICS-A | ICAES-A | dc, a | a |
| 8 | ICS-S | ICA-S | ICAS-S | dc | s |
| 9 | ICA-A | ICS-A | ICASA-A | dc | a |
| 10 | ICA-A | ICE-A | ICAEA-A | dc | a |

Table 7: List of compositional rules.

| Model | BW | EW | KC | LC | PB | | KB | |
|---|---|---|---|---|---|---|---|---|
| | *Acc* | *Acc* | *Acc* | *Acc* | *Bleu-2* | *R-L* | *Bleu-2* | *R-L* |
| ChatGPT (Zero-shot) | 75.2 | 31.3 | 79.3 | 62.9 | 0.95 | 7.3 | 6.65 | 15.22 |
| ChatGPT (One-shot) | 83.35 | 33.0 | 70.13 | 72.6 | 1.82 | 10 | 6.08 | 14.5 |

Table 8: Complementary evaluation results on atomic tasks — *c.f.*Table 4.

| Model | BW + EW | BW + KC | BW + LC | EW + KC | EW + LC | KC + LC | PB + EW | | EG + EW | |
|---|---|---|---|---|---|---|---|---|---|---|
| | Acc | Acc | Acc | Acc | Acc | Acc | EW Acc | R-L | EW Acc | R-L |
| ChatGPT (Zero-shot) | 30.8 | 65.4 | 50.4 | 18.3 | 9.05 | 27.9 | 13.3 | 9.37 | 45.7 | 54.3 |
| ChatGPT (One-shot) | 29.7 | 69.7 | 57.2 | 23.7 | 12.5 | 34.47 | 15.4 | 11.1 | 64.7 | 68.8 |

Table 9: Complementary evaluation results on compositional tasks — *c.f.*Table 5 *R-L* stands for *Rouge-L* metric, best results for each column are **bolded**.

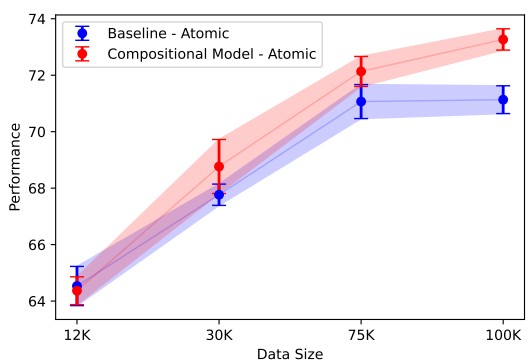

Figure 8: Atomic accuracy of both the baseline and compositional model over a varying number of training data sizes. Each datapoint is run across three indenpendently sampled test sets to account for variability

certain spurious traits in the datasets and in the way we preprocess our data. For example, for the *beginswith_generation* task, the evaluation checks if the given initial phrase and beginning of the response are exactly the same. During tokenization we split punctuation with an additional space *e.g.* ('The response should start with: Yes, I love this **song ,**') but ChatGPT omits the additional space while generating the response *i.e.* ('Yes, I love

| Model | # Shots | BW + EW | BW + KC | EW + KC |
|---|---|---|---|---|
| GPT-3.5 | 0 | 24.13 | 67.85 | 37.93 |
| GPT-3.5 | 3 | 31.03 | 78.57 | 44.82 |
| GPT-4 | 0 | 31.03 | 92.85 | 65.51 |
| GPT-4 | 3 | 62.06 | 86.71 | 75.86 |

Table 10: Evaluation results on three compositional tasks for a smaller test set using more incontext samples, and GPT-4 API.

**Instruction:** In this task you are given a dialog and an initial phrase. You need to generate a response which begins with the initial phrase.
**Context:**
**Speaker 1:** Have any plans for the weekend , Tom ?
**Speaker 2:** Yeah , I ' m going for a hike in the southern Rocky Mountains.
**Speaker 1:** Oh , do you go hiking often ?
**INITIAL PHRASE:** Not really,
Given this context generate a response that starts with the given initial sentence
**Answer:**

Table 11: Sample Prompt used for ChatGPT-based benchmarking

this **song,**'), and thus indirectly 'miss-generates' the correct response. Another simple mistake it does for *endswith_generation* is that it generates a sentence that actually ends with the given phrase, however, it does not stop the generation and adds another sentence to the response. *e.g.* for the task: 'The response should end with: **song.**', ChatGPT might generate: 'I like this **song**. Where can I listen to it?'.

To examine the impact of incorporating more examples in the context and employing enhanced proprietary models, we reassessed the outcomes for three tasks with a reduced test set, as seen in *c.f.* Table 10. Generally, GPT-4 performs notably better than GPT-3.5. Additionally, adding more contextual examples considerably boosts performance.

## E   Chain of Thought Potential in CESAR

We can extend CESAR to also incorporate reasoning supervision to the dialog tasks. This can be done by including Chain of thought elements into the dialog tasks. The CESAR task from Eq. (1) is modified as follows:

$$IC\Lambda - \Lambda'\psi =$$
$$IC\underbrace{\{g(\lambda_1),\ldots,g(\lambda_m)\}}_{\text{grounding}} - \underbrace{\{g(\lambda'_1),\ldots,g(\lambda'_n)\}}_{\text{reasoning or CoT}}\psi,$$

$$(2)$$

where, $\Lambda' = \{g(\lambda'_1),\ldots,\ g(\lambda'_n)\}$ is interpreted as the multiset of dialog components that can be used to reason about the primary output $\psi$. In the traditional CoT framework, the reasoning precedes the main output, thus we represent the output sequence as $\Lambda'\psi$.[7]

---

**Definition 3** (CoT-Generation)**:** *Any i-D Task, $IC\Lambda - \psi$ with $|\Lambda| = i$ and $i \geq 1$, can be converted to its CoT counterpart by shifting a subset of dialog items from input to output, i.e., $IC\left(\Lambda \setminus \{\lambda_k\}\right) - \left(\{\lambda_k\}\right)\psi$.*

---

Note that any $i$-D Task can be converted to any $j$-D Task where $j < i$ using Definition 3 repeatedly. For example, a CESAR Task $ICSSEA - R$ can be converted to $ICA - SSER$ after three iterations of Definition 3 in shifting $S, S, E$ dialog components.

These operations ensure full task coverage of any arbitrary CESAR task as per Eq. (1).

# F  ChatGPT Evaluation Prompts

Earlier studies have shows that ChatGPT does a good job in evaluating the overall quality of generated text by language models (Zheng et al., 2023). We used ChatGPT to evaluate the EW+EMG task in Table 6. Since emotion grounded generation is hard to evaluate we utilize ChatGPT and classify the emotions generated by each model and then calculate the accuracy of each model. Moreover because models may tend to generate the name of the emotion directly rather than infusing the emotion into the response (*e.g.* for response generation grounded on happiness the generated response can be 'I am very happy!") we further generate qualitative scores by ChatGPT for each of model's responses.

We use in-context examples to set each of these evaluation prompts. Table 12 and Table 13 depict template prompts for emotion classification and quality evaluation respectively.

---

[7]While CoT is accommodated in the CESAR framework, we do not keep it in the scope of our experiments. Thus, in our experiments, $\Lambda' = \{\phi\}$, i.e., $\Lambda'$ is an empty set.

---

You are a helpful assistant that helps evaluate a response given to a user query. Your job is to tell if the response answers the user's query.
Given the dialog:
Speaker 1: Hi Mark, I am going to the pool tomorrow, would you like to join?
Speaker 2: Hi Alice, sure thing, I was thinking my self to go recently.
Speaker 1: Great, Peter may join us too!
Speaker 2: Nice! It has been so long since I have seen Peter, would be great to see him.
Which of the following emotion does the final turn depict: happiness, fear, surprise, disgust, sadness, anger or no_emotion?
happiness
Given the dialog:
Speaker 1: What are you working on Sam?
Speaker 2: I am still trying to solve the math problem from last week, I think I am close.
Speaker 1: I see, I was able to solve it last night before going to bed.
Which of the following emotion does the final turn depict: fear, surprise, joy, neutral, disgust or sadness, anger? neutral
Given the dialog:
**<Dialog Context>**
Which of the following emotion does the final turn depict:
**<cand_emotions>**

Table 12: Prompt template used for ChatGPT to evaluate emotion-grounded generation

---

You are a helpful assistant that helps evaluate a response given to a user query. Your job is to tell if the response answers the user's query.
Given the dialog:
Speaker 1: Hi Mark, I am going to the pool tomorrow, would you like to join?
Speaker 2: Hi Alice, sure thing, I was thinking my self to go recently.
Speaker 1: Great, Peter may join us too!
How good is the response: "Nice! It has been so long since I have seen Peter, would be great to see him."?
Provide your evaluation as a score between 1 and 5, where 1 is for answers of bad quality, such as dull or irrelevant answers, and 5 is for answers that are relevant to the dialog and are not generic.
Score: 4/5.
Given the dialog:
Speaker 1: What are you working on Sam?
Speaker 2: I am still trying to solve the math problem from last week, I think I am close.
How good is the response: "I am going to bed."?
Provide your evaluation as a score between 1 and 5, where 1 is for answers of bad quality, such as dull or irrelevant answers, and 5 is for answers that are relevant to the dialog and are not generic.
Score: 1/5.
Given the dialog:
**<Dialog Context>**
How good is the response: "**<response>**"? Provide your evaluation as a score between 1 and 5, where 1 is for answers of bad quality, such as dull or irrelevant answers, and 5 is for answers that are relevant to the dialog and are not generic

Table 13: Prompt template used for ChatGPT to evaluate response quality.

**Instruction:** In this task you will generate an utterance given a dialogue context and an emotion and In this task, you will be shown a conversation context and a phrase. You need to generate a response to the conversation based on the context which ends with the provided phrase.
**Input:**
**Dialog Context** You'll never guess what I won at work today ! - Tickets to tonight's final NBA game.
**Emotion:** surprise
**Final Phrase:** planning on taking me !
The response with the given emotion is and Given this context and final phrase, the response is:

(a) Compositional task by *Naive Composer*

**Instruction:** Provide the correct value for response fields given the dialog context and action fields.
**Actions:**
The emotion of the next turn should be: surprise
The response should end with this sentence: planning on taking me !
**Dialog Context:**
Speaker 1: You'll never guess what I won at work today ! - Tickets to tonight's final NBA game.
**Response:**

(b) Compositional task by *CESAR*

Figure 9: Figure depicting naive composition vs CESAR composition.

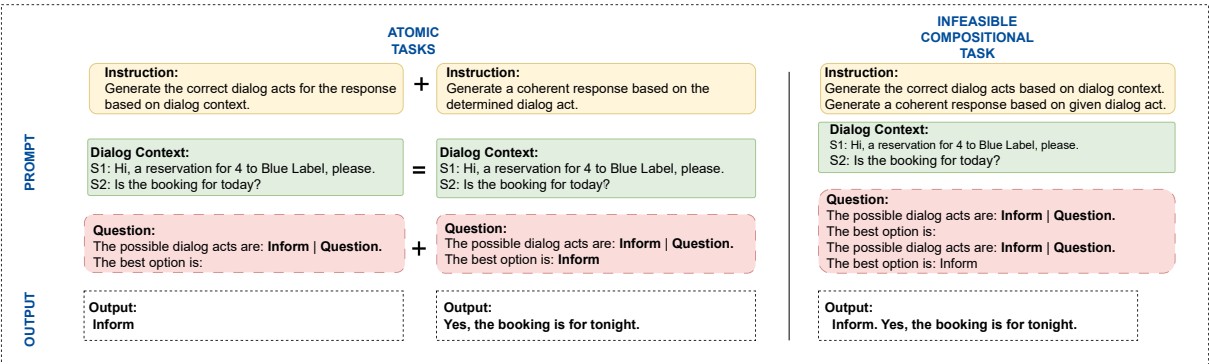

Figure 10: Infeasible Compositional Task example resulting from naive composition of atomic tasks.

## G   Can Naive Conjunctions provide Complex Instructions?

To get additional data for complex instructions, we could simply append single-task instructions and create complex ones. We use this method at Table 6 named as the *Naive Composer* to see how this compares to CESAR prompts. Fig. 9 shows two example prompts for the same task generated by the *Naive Composer* vs *CESAR*.

It is important to note that for these experiments we manually feed *Naive Composer* which tasks it should compose together. However, can we use this naive method to provide complex instructions at a scale? We claim that the answer to this question is No. In dialog tasks, such naive compositions of tasks could lead to either non-sensical/invalid tasks or create tasks that are very rare and not useful. Let us look at an example — Fig. 10. Here we have two atomic tasks on the left hand side, *dialog_act_prediction* and *dialog_act_grounded_generation*. If one were to implement a naive composer that would blindly combine two tasks whenever they have the same dialog context this would result in the infeasible

task on the right-hand side. This task is infeasible because the answer to first task is incorporated in the input of the second task.

## H   Structured vs. Natural Prompts

The structured form of prompts that CESAR framework employs makes it easy to scale both atomic and compositional tasks up. However, this comes with a potential caveat. One could argue that a model trained on CESAR formatted prompts might struggle to process natural human queries. In this section we discuss that the benefits that the CESAR framework brings can possibly be enjoyed without the structured form it imposes because the conversion between two is made easy by large language models, *e.g.* ChatGPT. We show an example of this conversion using ChatGPT where we provide an in-context example in Table 14. This example indicates that given an in-context example, ChatGPT model is able to convert a CESAR formatted prompt to a natural format that can be used in model training.

| Sections of the Prompt | Content |
|---|---|
| Instruction | Convert the following language model prompts from their structured format to a more natural format that is likely to be written by a human. Follow the exact structure in each example: |
| In context example | **Original Prompt:**
Instruction: Fill in the correct value for the action field given the dialog context.
Dialog Context:
A: Hi, how was your day at work
B: It went great, I had a very productive meeting.
A: Great, what tips would you give to get a productive day at work?
Action:
The dialog act of the next response should be:

**Natural Prompt:**
What should the dialog act of the next response be?
Dialog context:
Dialog Context:
A: Hi, how was your day at work
B: It went great, I had a very productive meeting.
A: Great, what tips would you give to get a productive day at work? |
| Query | **Original Prompt:**
Instruction: Fill in the correct value for the action field given the dialog context.
Dialog Context:
A: Hi, what is your favorite activity for week nights?
B: I like to listen to music while walking in the city.
A: Great, what band is your favorite?
Action:
The length of the next response should be:
**Natural Prompt:** |
| Output | What should be the length of the next response?
Dialog context:
A: Hi, what is your favorite activity for week nights?
B: I like to listen to music while walking in the city.
A: Great, what band is your favorite? |

Table 14: Example prompt and output pair, utilizing the ChatGPT model to convert CESAR formatted prompt to natural format. The first three rows are sections of the prompt provided to ChatGPT whereas the last row is the output generated by ChatGPT.

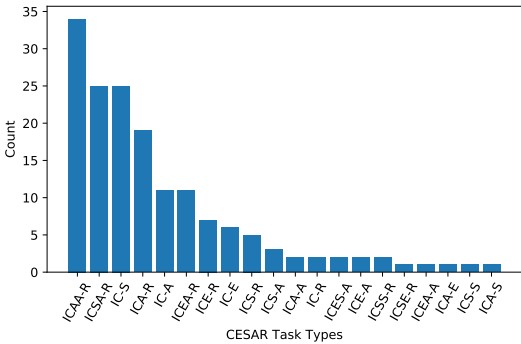

Figure 11: CESAR task distribution in InstructDial++
dataset

**Prompt:**
Instruction: Provide the correct value for response fields given the dialog context and action fields.
Input:
Dialog Context:
Speaker 1: That's a rough day for it. I had my heart broken once by a girl because her sister was jealous of us. Her sister claimed that she and I had a thing and my girlfriend (at the time) believed the lie.
Speaker 2: That's like rough day, the song .
Speaker 1: I don't know that song, who sings it?
Actions:
The final sentence of the response should be: "guy, named Paulini."
Response:
**Output:**
It's by a guy, named Paulini.

(a) Atomic Task example

**Prompt:**
Instruction: Provide the correct value for response fields given the dialog context and action fields.
Input:
Dialog Context:
Speaker 1: What kind of place shall we rent ?
Speaker 2: It should be close to the university . Neither of us are good at getting up in the mornings and closer it is , the later we can get up .
Actions:
The response should start with this initial phrase: "Absolutely . That's"
The response should contain the following keywords: "thing" and "flat"
Response:
**Output:**
Absolutely. That's the most important thing and flat should be furnished.

(b) Compositional Task example

Figure 12: Qualitative examples generated by the *CESAR* framework.

| Task Name | Datasets | CESAR Task |
|---|---|---|
| act_prediction* | MSRE2E (Li et al., 2018), MultiWOZ (Budzianowski et al., 2018), DailyDialog (Zhao et al., 2020), Curiosity-Dialogs* (Rodriguez et al., 2020), | IC-A |
| belief_state_prediction* | MultiWOZ (Budzianowski et al., 2018), KVRET (Eric et al., 2017), WOZ (Mrkšić et al., 2017), CAM-REST676 (Wen et al., 2017), MSRE2E (Li et al., 2018), Frames (El Asri et al., 2017), TaskMaster (Byrne et al., 2019), DSTC8-SGD (Rastogi et al., 2019) | IC-A |
| emotion_prediction* | EmotionLines (Hsu et al., 2018), GoEmotions (Demszky et al., 2020), DailyDialog (Zhao et al., 2020), FriendsED* (Zahiri and Choi, 2017) | IC-A |
| intent_prediction* | ATIS (Hemphill et al., 1990), SNIPS (Coucke et al., 2018), CLINC-150 (Larson et al., 2019), HWU64 (Liu et al., 2019), Banking77 (Casanueva et al., 2020), DiaSafety* (Sun et al., 2021) | IC-A |
| nli_prediction | Dialogue NLI (Welleck et al., 2019), DECODE (Nie et al., 2021) | IC-A |
| persuasion_prediction* | CaSiNo (Chawla et al., 2021), Persuasion (Wang et al., 2019) | IC-A |
| response_length_prediction* | DailyDialog (Zhao et al., 2020), WoW (Dinan et al., 2018), EmpatheticDialogues (Rashkin et al., 2019) | IC-A |
| slot_present_prediction | slot-dstc8_sgd, ATIS (Hemphill et al., 1990), SNIPS (Coucke et al., 2018), TaskMaster (Byrne et al., 2019), MSRE2E (Li et al., 2018) | IC-A |
| slot_value_prediction | slot-dstc8_sgd, ATIS (Hemphill et al., 1990), SNIPS (Coucke et al., 2018), TaskMaster (Byrne et al., 2019), MSRE2E (Li et al., 2018) | IC-A |
| speaker_prediction* | Molweni* (Li et al., 2020) | IC-A |
| keyword_prediction* | empathy (Sharma et al., 2020), DailyDialog (Zhao et al., 2020), CONVAI (Dinan et al., 2019b), EmotionLines (Hsu et al., 2018), WoW (Dinan et al., 2018) | IC-A |
| act_generation* | woz, MSRE2E (Li et al., 2018), MultiWOZ (Budzianowski et al., 2018), DailyDialog (Zhao et al., 2020), CuriosityDialogs* (Rodriguez et al., 2020) | ICA-R |
| advice_generation | TuringAdvice (Zellers et al., 2021) | ICA-R |
| beginswith_controlled_generation | empathy (Sharma et al., 2020), DailyDialog (Zhao et al., 2020), CONVAI (Dinan et al., 2019b), TuringAdvice (Zellers et al., 2021), EmotionLines (Hsu et al., 2018), WoW (Dinan et al., 2018) | ICA-R |
| db_based_generation | MultiWOZ (Budzianowski et al., 2018) | ICA-R |
| edit_generation | TopicalChat (Gopalakrishnan et al., 2019), EmotionLines (Hsu et al., 2018), Persuasion (Wang et al., 2019), CaSiNo (Chawla et al., 2021), DialogSum (Chen et al., 2021), empathy (Sharma et al., 2020), DailyDialog (Zhao et al., 2020), CONVAI (Dinan et al., 2019b), EmotionLines (Hsu et al., 2018), WoW (Dinan et al., 2018) | ICA-R |
| emotion_generation | GoEmotions (Demszky et al., 2020), EmotionLines (Hsu et al., 2018), DailyDialog (Zhao et al., 2020), FriendsED* (Zahiri and Choi, 2017) | ICA-R |
| endswith_controlled_generation | empathy (Sharma et al., 2020), DailyDialog (Zhao et al., 2020), CONVAI (Dinan et al., 2019b), TuringAdvice (Zellers et al., 2021), EmotionLines (Hsu et al., 2018), WoW (Dinan et al., 2018) | ICA-R |
| intent_generation* | ATIS (Hemphill et al., 1990), SNIPS (Coucke et al., 2018), CLINC-150 (Larson et al., 2019), HWU64 (Liu et al., 2019), Banking77 (Casanueva et al., 2020), DiaSafety* (Sun et al., 2021) | ICA-R |
| keyword_controlled_generation | empathy (Sharma et al., 2020), DailyDialog (Zhao et al., 2020), CONVAI (Dinan et al., 2019b), EmotionLines (Hsu et al., 2018), WoW (Dinan et al., 2018) | ICA-R |
| nli_generation* | Dialogue NLI (Welleck et al., 2019), DECODE (Nie et al., 2021) | ICA-R |
| nontoxic_feedback_generation* | SaFeRDialogues (Ung et al., 2022) | ICA-R |
| persuasion_generation | CaSiNo (Chawla et al., 2021), Persuasion (Wang et al., 2019) | ICA-R |
| recovery_generation | SaFeRDialogues (Ung et al., 2022) | ICA-R |
| response_generation_length* | DailyDialog (Zhao et al., 2020), WoW (Dinan et al., 2018), EmpatheticDialogues (Rashkin et al., 2019) | ICA-R |
| schema_based_generation | FloDial (Raghu et al., 2021) | ICA-R |
| slot_present_generation* | DSTC8-SGD (Rastogi et al., 2019), ATIS (Hemphill et al., 1990), SNIPS (Coucke et al., 2018), TaskMaster (Byrne et al., 2019), MSRE2E (Li et al., 2018) | ICA-R |
| slot_value_grounded_generation* | slot-dstc8_sgd, ATIS (Hemphill et al., 1990), SNIPS (Coucke et al., 2018), TaskMaster (Byrne et al., 2019), MSRE2E (Li et al., 2018) | ICA-R |

| Task Name | Datasets | CESAR Task |
|---|---|---|
| speaker_generation* | Molweni* (Li et al., 2020) | ICA-R |
| target_controlled_generation | OTTERS (Sevegnani et al., 2021) | ICA-R |
| dialfact_classification | DialFact (Gupta et al., 2021) | IC-E |
| persona_generation* | CONVAI (Dinan et al., 2019b), PersonaChat (Zhang et al., 2018), FriendsPD* (Jiang et al., 2019) | IC-E |
| answer_evidence_generation* | MuTual (Cui et al., 2020), CoQA (Reddy et al., 2018), QuAC (Choi et al., 2018), CIDER (Ghosal et al., 2021), WikiDial* (Dai et al., 2022) | IC-E |
| document_generation* | doc2dial (Feng et al., 2020) | IC-E |
| graph_generation* | OpenDialKG (Moon et al., 2019) | IC-E |
| knowledge_generation* | TopicalChat (Gopalakrishnan et al., 2019), WoW (Dinan et al., 2018) | IC-E |
| answer_generation | MuTual (Cui et al., 2020), CoQA (Reddy et al., 2018), QuAC (Choi et al., 2018), CIDER (Ghosal et al., 2021), WikiDial* (Dai et al., 2022) | ICE-R |
| answer_selection | CoQA (Reddy et al., 2018), QuAC (Choi et al., 2018), MuTual (Cui et al., 2020), CIDER (Ghosal et al., 2021) | ICE-R |
| commonsense_grounded_generation* | Soda* (Kim et al., 2022), Commonsense* (Zhou et al., 2021) | ICE-R |
| document_grounded_generation | doc2dial (Feng et al., 2020) | ICE-R |
| graph_based_generation | OpenDialKG (Moon et al., 2019) | ICE-R |
| knowledge_grounded_generation | TopicalChat (Gopalakrishnan et al., 2019), WoW (Dinan et al., 2018) | ICE-R |
| persona_grounded_generation | CONVAI (Dinan et al., 2019b), PersonaChat (Zhang et al., 2018), FriendsPD* (Jiang et al., 2019) | ICE-R |
| eval_ranking | USR (Mehri and Eskenazi, 2020b), FED(Mehri and Eskenazi, 2020a), GRADE (Huang et al., 2020), HU-MOD (Merdivan et al., 2020), Anthropic* (Bai et al., 2022) | IC-R |
| response_generation | DailyDialog (Zhao et al., 2020), CONVAI (Dinan et al., 2019b), WoW (Dinan et al., 2018), EmpatheticDialogues (Rashkin et al., 2019), OpenDialKG (Moon et al., 2019), Anthropic* (Bai et al., 2022), Multitalk* (Dou et al., 2021) | IC-R |
| act_classification | MSRE2E (Li et al., 2018), MultiWOZ (Budzianowski et al., 2018), DailyDialog (Zhao et al., 2020), CuriosityDialogs* (Rodriguez et al., 2020) | IC-S |
| advice_present | TuringAdvice (Zellers et al., 2021) | IC-S |
| belief_state_generation | MultiWOZ (Budzianowski et al., 2018), KVRET (Eric et al., 2017), WOZ (Mrkšić et al., 2017), CAM-REST676 (Wen et al., 2017), MSRE2E (Li et al., 2018), Frames (El Asri et al., 2017), TaskMaster (Byrne et al., 2019), DSTC8-SGD (Rastogi et al., 2019) | IC-S |
| deal_present | Deal (Lewis et al., 2017) | IC-S |
| discourse_parsing* | CIDER (Ghosal et al., 2021), Molweni* (Li et al., 2020) | IC-S |
| emotion_tagging | EmotionLines (Hsu et al., 2018), GoEmotions (Demszky et al., 2020), DailyDialog (Zhao et al., 2020), FriendsED* (Zahiri and Choi, 2017) | IC-S |
| eval_binary | USR (Mehri and Eskenazi, 2020b), FED(Mehri and Eskenazi, 2020a), GRADE (Huang et al., 2020), HU-MOD (Merdivan et al., 2020) | IC-S |
| eval_rating | USR (Mehri and Eskenazi, 2020b), FED(Mehri and Eskenazi, 2020a), GRADE (Huang et al., 2020), HU-MOD (Merdivan et al., 2020), ConTurE* (Ghazarian et al., 2022) | IC-S |
| find_incoherent_utterance | DailyDialog (Zhao et al., 2020), WoW (Dinan et al., 2018), EmpatheticDialogues (Rashkin et al., 2019), OpenDialKG (Moon et al., 2019) | IC-S |
| find_missing_utterance | DailyDialog (Zhao et al., 2020), CONVAI (Dinan et al., 2019b), WoW (Dinan et al., 2018), EmpatheticDialogues (Rashkin et al., 2019), OpenDialKG (Moon et al., 2019) | IC-S |
| find_swapped_utterance | DailyDialog (Zhao et al., 2020), CONVAI (Dinan et al., 2019b), WoW (Dinan et al., 2018), EmpatheticDialogues (Rashkin et al., 2019), OpenDialKG (Moon et al., 2019) | IC-S |
| intent_classification | ATIS (Hemphill et al., 1990), SNIPS (Coucke et al., 2018), CLINC-150 (Larson et al., 2019), HWU64 (Liu et al., 2019), Banking77 (Casanueva et al., 2020), DiaSafety* (Sun et al., 2021) | IC-S |
| intent_present | ATIS (Hemphill et al., 1990), SNIPS (Coucke et al., 2018), CLINC-150 (Larson et al., 2019), HWU64 (Liu et al., 2019), Banking77 (Casanueva et al., 2020) | IC-S |

| Task Name | Datasets | CESAR Task |
|---|---|---|
| nli_classification* | Dialogue NLI (Welleck et al., 2019), DECODE (Nie et al., 2021) | IC-S |
| person_identification* | FriendsRC* (Ma et al., 2018) | IC-S |
| persuasion_present | CaSiNo (Chawla et al., 2021), Persuasion (Wang et al., 2019) | IC-S |
| question_answering | FriendsQA* (Yang and Choi, 2019), CICERO* (Ghosal et al., 2022), Dream* (Sun et al., 2019) | IC-S |
| relation_classification | DialogRE (Yu et al., 2020) | IC-S |
| relation_present | DialogRE (Yu et al., 2020) | IC-S |
| response_length_classification | DailyDialog (Zhao et al., 2020), WoW (Dinan et al., 2018), EmpatheticDialogues (Rashkin et al., 2019) | IC-S |
| slot_present* | slot-dstc8_sgd, ATIS (Hemphill et al., 1990), SNIPS (Coucke et al., 2018), TaskMaster (Byrne et al., 2019), MSRE2E (Li et al., 2018) | IC-S |
| slot_tagging | slot-dstc8_sgd, ATIS (Hemphill et al., 1990), SNIPS (Coucke et al., 2018), TaskMaster (Byrne et al., 2019), MSRE2E (Li et al., 2018) | IC-S |
| speaker_selection* | Molweni* (Li et al., 2020) | IC-S |
| summarization | samsum, DialogSum (Chen et al., 2021), Soda* (Kim et al., 2022) | IC-S |
| toxic_classification | ToxiChat (Baheti et al., 2021), BAD (Xu et al., 2021), Build it Break it Fix it (Dinan et al., 2019a), DiaSafety* (Sun et al., 2021) | IC-S |
| act_classified_generation* | DailyDialog (Zhao et al., 2020), CuriosityDialogs* (Rodriguez et al., 2020) | ICS-R |
| response_length_classified_generation* | DailyDialog (Zhao et al., 2020), WoW (Dinan et al., 2018), EmpatheticDialogues (Rashkin et al., 2019) | ICS-R |
| persuasion_classified_generation* | CaSiNo (Chawla et al., 2021), Persuasion (Wang et al., 2019) | ICS-R |
| emotion_classified_generation* | DailyDialog (Zhao et al., 2020), FriendsED* (Zahiri and Choi, 2017) | ICS-R |
| belief_state_classified_generation* | CAMREST676 (Wen et al., 2017) | ICS-R |
| act_grounded_emotion_tagging* | DailyDialog (Zhao et al., 2020) | ICS-S |
| act_grounded_emotion_prediction* | DailyDialog (Zhao et al., 2020) | ICS-A |
| length_grounded_act_prediction* | MSRE2E (Li et al., 2018), MultiWOZ (Budzianowski et al., 2018), DailyDialog (Zhao et al., 2020), CuriosityDialogs* (Rodriguez et al., 2020) | ICS-A |
| length_grounded_keyword_prediction* | CONVAI (Dinan et al., 2019b), PersonaChat (Zhang et al., 2018), FriendsPD* (Jiang et al., 2019), MultiWOZ (Budzianowski et al., 2018), CuriosityDialogs* (Rodriguez et al., 2020) | ICS-A |
| emotion_predicted_act_classification* | DailyDialog (Zhao et al., 2020) | ICA-S |
| beginswith_grounded_endswith_prediction* | empathy (Sharma et al., 2020), DailyDialog (Zhao et al., 2020), CONVAI (Dinan et al., 2019b), TuringAdvice (Zellers et al., 2021), EmotionLines (Hsu et al., 2018), WoW (Dinan et al., 2018) | ICA-A |
| beginswith_grounded_length_prediction* | CONVAI (Dinan et al., 2019b), PersonaChat (Zhang et al., 2018), FriendsPD* (Jiang et al., 2019) | ICA-A |
| keyword_grounded_persona_generation* | CONVAI (Dinan et al., 2019b), PersonaChat (Zhang et al., 2018), FriendsPD* (Jiang et al., 2019) | ICA-E |
| persona_grounded_keyword_prediction* | CONVAI (Dinan et al., 2019b), PersonaChat (Zhang et al., 2018), FriendsPD* (Jiang et al., 2019) | ICE-A |
| persona_grounded_length_prediction* | CONVAI (Dinan et al., 2019b), PersonaChat (Zhang et al., 2018), FriendsPD* (Jiang et al., 2019) | ICE-A |

Table 15: CESAR downstream atomic tasks incorporating 0D and 1D grounding tasks. Tasks marked with a '*' are novel compared to the InstructDial benchmark (The table starts in the previous pages).

| Task Name | Datasets | CESAR Task |
|---|---|---|
| act_generation
+
edit_generation | DailyDialog (Zhao et al., 2020) | ICAA-R |
| act_generation
+
emotion_generation | DailyDialog (Zhao et al., 2020) | ICAA-R |
| act_generation
+
endswith_controlled_generation | DailyDialog (Zhao et al., 2020) | ICAA-R |
| act_generation
+
keyword_controlled_generation | DailyDialog (Zhao et al., 2020) | ICAA-R |
| act_generation
+
response_generation_length | DailyDialog (Zhao et al., 2020) | ICAA-R |
| act_generation
+
slot_present_generation | MSRE2E (Li et al., 2018) | ICAA-R |
| act_generation
+
slot_value_grounded_generation | MSRE2E (Li et al., 2018) | ICAA-R |
| advice_generation
+
beginswith_controlled_generation | TuringAdvice (Zellers et al., 2021) | ICAA-R |
| advice_generation
+
endswith_controlled_generation | TuringAdvice (Zellers et al., 2021) | ICAA-R |
| beginswith_controlled_generation
+
act_generation | DailyDialog (Zhao et al., 2020) | ICAA-R |
| beginswith_controlled_generation
+
emotion_generation | DailyDialog (Zhao et al., 2020), Emotion-Lines (Hsu et al., 2018) | ICAA-R |
| beginswith_controlled_generation
+
keyword_controlled_generation | CONVAI (Dinan et al., 2019b), DailyDia-log (Zhao et al., 2020), EmotionLines (Hsu et al., 2018), empathy (Sharma et al., 2020), WoW (Di-nan et al., 2018) | ICAA-R |
| edit_generation
+
beginswith_controlled_generation | CONVAI (Dinan et al., 2019b), DailyDia-log (Zhao et al., 2020), EmotionLines (Hsu et al., 2018), empathy (Sharma et al., 2020), WoW (Di-nan et al., 2018) | ICAA-R |
| edit_generation
+
emotion_generation | DailyDialog (Zhao et al., 2020), Emotion-Lines (Hsu et al., 2018) | ICAA-R |
| edit_generation
+
endswith_controlled_generation | CONVAI (Dinan et al., 2019b), DailyDia-log (Zhao et al., 2020), EmotionLines (Hsu et al., 2018), empathy (Sharma et al., 2020), WoW (Di-nan et al., 2018) | ICAA-R |
| edit_generation
+
keyword_controlled_generation | CONVAI (Dinan et al., 2019b), DailyDia-log (Zhao et al., 2020), EmotionLines (Hsu et al., 2018), empathy (Sharma et al., 2020), WoW (Di-nan et al., 2018) | ICAA-R |
| edit_generation
+
persuasion_generation | CaSiNo (Chawla et al., 2021), Persuasion (Wang et al., 2019) | ICAA-R |
| edit_generation
+
response_generation_length | DailyDialog (Zhao et al., 2020), WoW (Dinan et al., 2018) | ICAA-R |
| emotion_generation
+
endswith_controlled_generation | DailyDialog (Zhao et al., 2020), Emotion-Lines (Hsu et al., 2018) | ICAA-R |
| emotion_generation
+
keyword_controlled_generation | DailyDialog (Zhao et al., 2020), Emotion-Lines (Hsu et al., 2018) | ICAA-R |

| Task Name | Datasets | CESAR Task |
|---|---|---|
| endswith_controlled_generation + beginswith_controlled_generation | CONVAI (Dinan et al., 2019b), DailyDialog (Zhao et al., 2020), EmotionLines (Hsu et al., 2018), empathy (Sharma et al., 2020), TuringAdvice (Zellers et al., 2021), WoW (Dinan et al., 2018) | ICAA-R |
| endswith_controlled_generation + emotion_classified_generation | DailyDialog (Zhao et al., 2020) | ICAA-R |
| endswith_controlled_generation + keyword_controlled_generation | CONVAI (Dinan et al., 2019b), DailyDialog (Zhao et al., 2020), EmotionLines (Hsu et al., 2018), empathy (Sharma et al., 2020), WoW (Dinan et al., 2018) | ICAA-R |
| nontoxic_feedback_generation + recovery_generation | SaFeRDialogues (Ung et al., 2022) | ICAA-R |
| response_generation_length + beginswith_controlled_generation | DailyDialog (Zhao et al., 2020), WoW (Dinan et al., 2018) | ICAA-R |
| response_generation_length + emotion_classified_generation | DailyDialog (Zhao et al., 2020) | ICAA-R |
| response_generation_length + emotion_generation | DailyDialog (Zhao et al., 2020) | ICAA-R |
| response_generation_length + endswith_controlled_generation | DailyDialog (Zhao et al., 2020), WoW (Dinan et al., 2018) | ICAA-R |
| response_generation_length + keyword_controlled_generation | DailyDialog (Zhao et al., 2020), WoW (Dinan et al., 2018) | ICAA-R |
| slot_present_generation + slot_value_grounded_generation | MSRE2E (Li et al., 2018), slot-dstc8_sgd | ICAA-R |
| act_classified_generation + emotion_classified_generation | DailyDialog (Zhao et al., 2020) | ICSS-R |
| act_classified_generation + response_length_classified_generation | DailyDialog (Zhao et al., 2020) | ICSS-R |
| persona_grounded_length_prediction + beginswith_grounded_length_prediction | CONVAI (Dinan et al., 2019b), PersonaChat (Zhang et al., 2018) | ICEA-A |
| act_generation + db_based_generation | MultiWOZ (Budzianowski et al., 2018) | ICEA-R |
| edit_generation + knowledge_grounded_generation | TopicalChat (Gopalakrishnan et al., 2019), WoW (Dinan et al., 2018) | ICEA-R |
| edit_generation + persona_grounded_generation | CONVAI (Dinan et al., 2019b) | ICEA-R |
| endswith_controlled_generation + persona_grounded_generation | CONVAI (Dinan et al., 2019b) | ICEA-R |
| knowledge_grounded_generation + beginswith_controlled_generation | WoW (Dinan et al., 2018) | ICEA-R |
| knowledge_grounded_generation + endswith_controlled_generation | WoW (Dinan et al., 2018) | ICEA-R |
| knowledge_grounded_generation + keyword_controlled_generation | WoW (Dinan et al., 2018) | ICEA-R |
| persona_grounded_generation + beginswith_controlled_generation | CONVAI (Dinan et al., 2019b) | ICEA-R |

| Task Name | Datasets | CESAR Task |
|---|---|---|
| persona_grounded_generation
+
keyword_controlled_generation | CONVAI (Dinan et al., 2019b) | ICEA-R |
| response_generation_length
+
knowledge_grounded_generation | WoW (Dinan et al., 2018) | ICEA-R |
| persona_grounded_keyword_prediction
+
length_grounded_keyword_prediction | CONVAI (Dinan et al., 2019b), PersonaChat (Zhang et al., 2018) | ICES-A |
| act_classified_generation
+
emotion_generation | DailyDialog (Zhao et al., 2020) | ICSA-R |
| act_classified_generation
+
endswith_controlled_generation | DailyDialog (Zhao et al., 2020) | ICSA-R |
| act_classified_generation
+
keyword_controlled_generation | DailyDialog (Zhao et al., 2020) | ICSA-R |
| act_generation
+
act_classified_generation | CuriosityDialogs* (Rodriguez et al., 2020), DailyDialog (Zhao et al., 2020) | ICSA-R |
| act_generation
+
emotion_classified_generation | DailyDialog (Zhao et al., 2020) | ICSA-R |
| act_generation
+
response_length_classified_generation | DailyDialog (Zhao et al., 2020) | ICSA-R |
| beginswith_controlled_generation
+
act_classified_generation | DailyDialog (Zhao et al., 2020) | ICSA-R |
| beginswith_controlled_generation
+
emotion_classified_generation | DailyDialog (Zhao et al., 2020) | ICSA-R |
| edit_generation
+
act_classified_generation | DailyDialog (Zhao et al., 2020) | ICSA-R |
| edit_generation
+
emotion_classified_generation | DailyDialog (Zhao et al., 2020) | ICSA-R |
| edit_generation
+
response_length_classified_generation | DailyDialog (Zhao et al., 2020), WoW (Dinan et al., 2018) | ICSA-R |
| emotion_generation
+
emotion_classified_generation | DailyDialog (Zhao et al., 2020), FriendsED* (Zahiri and Choi, 2017) | ICSA-R |
| endswith_controlled_generation
+
response_length_classified_generation | DailyDialog (Zhao et al., 2020), WoW (Dinan et al., 2018) | ICSA-R |
| keyword_controlled_generation
+
emotion_classified_generation | DailyDialog (Zhao et al., 2020) | ICSA-R |
| Persuasion_classified_generation
+
edit_generation | CaSiNo (Chawla et al., 2021), Persuasion (Wang et al., 2019) | ICSA-R |
| Persuasion_classified_generation
+
Persuasion_generation | CaSiNo (Chawla et al., 2021), Persuasion (Wang et al., 2019) | ICSA-R |
| response_generation_length
+
act_classified_generation | DailyDialog (Zhao et al., 2020) | ICSA-R |

| Task Name | Datasets | CESAR Task |
|---|---|---|
| response_generation_length + response_length_classified_generation | DailyDialog (Zhao et al., 2020), EmpatheticDialogues (Rashkin et al., 2019), WoW (Dinan et al., 2018) | ICSA-R |
| response_length_classified_generation + beginswith_controlled_generation | DailyDialog (Zhao et al., 2020), WoW (Dinan et al., 2018) | ICSA-R |
| response_length_classified_generation + emotion_classified_generation | DailyDialog (Zhao et al., 2020) | ICSA-R |
| response_length_classified_generation + emotion_generation | DailyDialog (Zhao et al., 2020) | ICSA-R |
| response_length_classified_generation + keyword_controlled_generation | DailyDialog (Zhao et al., 2020), WoW (Dinan et al., 2018) | ICSA-R |
| knowledge_grounded_generation + response_length_classified_generation | WoW (Dinan et al., 2018) | ICSE-R |

Table 16: CESAR downstream compositional 2D tasks (The table starts in the previous pages).