# OpenReview forum: "CESAR: Automatic Induction of Compositional Instructions for Multi-turn Dialogs"
_EMNLP/2023/Conference — EMNLP 2023 Main_

### Official Review · Reviewer_5Uzf · 2023-08-05

**Soundness:** 3

**Excitement:**

3: Ambivalent: It has merits (e.g., it reports state-of-the-art results, the idea is nice), but there are key weaknesses (e.g., it describes incremental work), and it can significantly benefit from another round of revision. However, I won't object to accepting it if my co-reviewers champion it.

**Paper Topic And Main Contributions:**

This paper proposes a novel framework called CESAR for automatic induction of compositional instructions for multi-turn dialogs. The authors apply CESAR on the InstructDial benchmark and demonstrate its scalability in providing rich instructions. They also conduct experiments to evaluate the performance of CESAR on atomic and compositional tasks. The paper provides valuable contributions to the field of dialog systems and instruction-based multitasking.

**Questions For The Authors:**

1) Could you provide more detailed explanations of the CESAR framework in the paper? It would be helpful for readers to understand the methodology and implementation.
2) Can you provide more thorough evaluation and comparisons with existing methods or baselines in the field? This would strengthen the paper's contributions and provide more insights for readers.
3) Is it possible to provide implementation details or make the code available for reproducibility? This would greatly benefit the research community.


**Reasons To Accept:**

1) The paper introduces a novel framework, CESAR, which addresses the challenge of generating complex instructions for dialog tasks.
2) CESAR unifies a large number of dialog tasks in the same format and allows programmatic induction of complex instructions without any manual effort.
3) The paper enhances the InstructDial benchmark with new datasets and tasks and demonstrates the scalability of CESAR in providing rich instructions.

**Reasons To Reject:**

1) The paper lacks a detailed explanation of the CESAR framework, making it difficult for readers to fully understand the methodology.
2) The evaluation section could benefit from more detailed analysis and comparisons with existing methods or baselines.
3) The paper does not provide implementation details or code availability, which could hinder reproducibility.

**Reproducibility:**

3: Could reproduce the results with some difficulty. The settings of parameters are underspecified or subjectively determined; the training/evaluation data are not widely available.

**Reviewer Confidence:**

3: Pretty sure, but there's a chance I missed something. Although I have a good feel for this area in general, I did not carefully check the paper's details, e.g., the math, experimental design, or novelty.

---

> ### Author Rebuttal · Authors · 2023-08-28
>
> We appreciate the detailed feedback from Reviewer "5Uzf". It is motivating to hear that they think **CESAR is a novel framework** and we are happy that they see value in its **capability to unify a large number of dialog tasks**, **generate complex instructions** and its **enhancements on the InstructDial benchmark**.
>
> CESAR centers on the generation of prompts and instructions within dialogues. Our principal innovation lies in its capacity to craft compositional task prompts, an aspect previously overlooked in prior studies in this domain.
>
> Below, we address each of Reviewer's concerns:
>
> > The paper lacks a detailed explanation of the CESAR framework, making it difficult for readers to fully understand the methodology.
>
> > Could you provide more detailed explanations of the CESAR framework in the paper? It would be helpful for readers to understand the methodology and implementation.
>
> Thank you for your feedback. We aimed to provide a comprehensive explanation in the paper. To elaborate:
> - Section 4.1 delves into the overarching framework and the structure of the prompts within CESAR.
> - In Section 4.2, we elucidate how 0-D, 1-D, and 2-D CESAR tasks are constructed, highlighting the nuances between atomic and compositional tasks.
> - Furthermore, Section 5.2 describes the mapping of each task in InstructDial++ to a CESAR task.
> - We genuinely value your perspective and would be grateful for any specific suggestions or points where you feel more clarity could be added.
>
> > The evaluation section could benefit from more detailed analysis and comparisons with existing methods or baselines.
>
> > Can you provide more thorough evaluation and comparisons with existing methods or baselines in the field? This would strengthen the paper's contributions and provide more insights for readers.
>
> - We have endeavored to provide a detailed explanation of both atomic and compositional results in the evaluation section.
> - Our evaluation is broadened through generalization experiments that span three dimensions. This includes evaluations of CESAR's efficacy with both seen and unseen compositions, as well as those involving new tasks.
> - We believe that these multifaceted discussions together offer solid backing for our hypothesis, further strengthened by the insights from our preliminary experiments.
> - We would appreciate any specific insights or suggestions the reviewer might have to enhance our paper.
>
> > The paper does not provide implementation details or code availability, which could hinder reproducibility.
>
> > Is it possible to provide implementation details or make the code available for reproducibility? This would greatly benefit the research community.
>
> - As mentioned in the abstract we will publish the implementation publicly upon acceptance.

---

### Official Review · Reviewer_8BXP · 2023-08-10

**Soundness:** 2

**Excitement:**

4: Strong: This paper deepens the understanding of some phenomenon or lowers the barriers to an existing research direction.

**Paper Topic And Main Contributions:**

The key contribution of this paper is proposing CESAR and InstructDial++, a framework to benchmark model's capability to handle compositional tasks.

CESAR is able to unify multiple compositional task dataset.

CESAR enables automatic generation of compositional dialog tasks.

The paper also make evaluation pipeline for compositional tasks.

**Questions For The Authors:**

Could you please show the assumption "open source models are really less capable of handling compositional tasks" is correct?

Could you please explain why "ChatGPT does not perform well on author's generated compositional task compared to the finetuned model".

Could you please further include evaluation on more open source models like Alpaca, Vicuna, and WizardLM, Llama2-Chat. Please compare their performance in a single table.

Could you please explain the correctness of CESAR, is it different from natural queries? Could you please filter some data collected from ShareGPT or somewhere else which has compositional task instructions as another benchmark? I doubt that your dataset is "aritificial".

**Reasons To Accept:**

The paper proposed a new form to represent compositional tasks.

The paper created a new benchmark InstructDial++ to evaluate model's capability on compositional tasks.

**Reasons To Reject:**

Firstly, I suspect that the assumption of this paper is wrong. Are open source models really less capable of handling compositional tasks? I did not find any evaluation on current state-of-the-art Llama models in the paper.

Secondly, I saw in this paper (Table 7) that ChatGPT does not perform well on author's generated compositional task compared to the finetuned model (Table 5). It is likely that the author created an "artful" dataset.

**Reproducibility:**

4: Could mostly reproduce the results, but there may be some variation because of sample variance or minor variations in their interpretation of the protocol or method.

**Reviewer Confidence:**

3: Pretty sure, but there's a chance I missed something. Although I have a good feel for this area in general, I did not carefully check the paper's details, e.g., the math, experimental design, or novelty.

**Typos Grammar Style And Presentation Improvements:**

Currently, the evaluation for ChatGPT is placed in Appendix, while the results on author's finetuned model is in main section. They should be placed together to show the author's assumtion is correct. And results from more open source models should be included. Otherwise I could not verify the soundness of this paper.

---

> ### Author Rebuttal · Authors · 2023-08-28
>
> We appreciate the detailed feedback from Reviewer "8BXP" and thank them for their encouraging views on **the novelty of our work** and the value they see in **InstructDial++ benchmark**.
>
> CESAR centers on the generation of prompts and instructions within dialogues. Our principal innovation lies in its capacity to craft compositional task prompts, an aspect previously overlooked in prior studies in this domain.
>
> Below, we address each of the Reviewer's concerns:
>
> > Firstly, I suspect that the assumption of this paper is wrong. Are open-source models really less capable of handling compositional tasks? I did not find any evaluation on current state-of-the-art Llama models in the paper.
>
> > Could you please show the assumption "open source models are really less capable of handling compositional tasks" is correct?
>
> > Could you please further include evaluation on more open source models like Alpaca, Vicuna, and WizardLM, Llama2-Chat. Please compare their performance in a single table.
>
>
> | Model        | Release Date | Contemporaneous according to ACL policy? |
> |--------------|--------------|------------------------------------------|
> | [Llama](https://ai.meta.com/blog/large-language-model-llama-meta-ai/)        | 24.02.2023   | YES                                      |
> | [Alpaca](https://crfm.stanford.edu/2023/03/13/alpaca.html)       | 13.03.2023   | YES                                      |
> | [Vicuna](https://lmsys.org/blog/2023-03-30-vicuna/)       | 30.03.2023   | YES                                      |
> | [WizardLM](https://github.com/nlpxucan/WizardLM)    | 09.08.2023   | YES                                      |
> | [Llama2-Chat](https://huggingface.co/blog/llama2)  | 18.07.2023   | YES                                      |
>
>
> - As presented all SOTA models at the time were released approximately in the last three months before our submission date which can be tagged as "contemporaneous" as per [ACL policy](https://www.aclweb.org/adminwiki/index.php?title=ACL_Policies_for_Submission,_Review_and_Citation)
> - Our experimentation involved utilizing the contemporary state-of-the-art model at that time, namely Flan-T5. Flan-T5 is a good foundation model and competes with many of these latest Llama derivatives (https://arxiv.org/abs/2306.04757).
> - However, we also acknowledge the reviewer's suggestion that incorporating results on these models is pertinent due to the dynamically changing nature of the field. Thus, we shall incorporate the latest instruction-tuned models in our revised experiments.
>
> > Secondly, I saw in this paper (Table 7) that ChatGPT does not perform well on the author's generated compositional task compared to the finetuned model (Table 5). It is likely that the author created an "artful" dataset.”
>
> > Could you please explain why "ChatGPT does not perform well on author's generated compositional task compared to the finetuned model".
>
> > Could you please explain the correctness of CESAR, is it different from natural queries? Could you please filter some data collected from ShareGPT or somewhere else that has compositional task instructions as another benchmark? I doubt that your dataset is "artificial".
>
> - We would like to highlight that CESAR is not composed of natural queries but of structured queries that enable the automatic combination of tasks (which is the core contribution of this work).
> - We discuss the advantages and disadvantages of this trait in Appendix G and also discuss ways to translate this structure into natural form using an LLM such as ChatGPT. We also show such an example in Table 11.
> - Moreover, we would like to bring another recent paper to the reviewer's attention which induces compositional prompts structurally (via CFG grammar) and later translates them to natural prompts → https://arxiv.org/abs/2307.08689
> - The results in Table 7 are poor as ChatGPT was prompted in zero-shot. In order to explore this further we experimented with a small set with 3 compositional tasks incorporating both GPT3.5 and GPT4 and both zero-shot and three-shot prompting. The results are as follows:
>
> | Model  | # SHOTS | BW+EW  | BW+KC | EW+KC |
> |--------|---------|--------|-------|-------|
> | GPT-3.5| 0       | 24.13  | 67.85 | 37.93 |
> | GPT-3.5| 3       | 31.03  | 78.57 | 44.82 |
> | GPT-4  | 0       | 31.03  | 92.85 | 65.51 |
> | GPT-4  | 3       | 62.06  | 85.71 | 75.86 |
>
> - In general, we find that with few examples in the context and more recently available GPT-4 API, the results improve significantly. In the final version of the paper, we shall update Table 7 with the above setting to make the distinction between open and closed models clearer.

---

### Official Review · Reviewer_dek5 · 2023-08-11

**Soundness:** 2

**Excitement:**

3: Ambivalent: It has merits (e.g., it reports state-of-the-art results, the idea is nice), but there are key weaknesses (e.g., it describes incremental work), and it can significantly benefit from another round of revision. However, I won't object to accepting it if my co-reviewers champion it.

**Paper Topic And Main Contributions:**

The paper presents CESAR, an innovative instruction-based framework designed for dialogue tasks. Unique in its capability, CESAR autonomously generates compositional task prompts, eliminating the need for human intervention. Building on this, the study expands the existing InstructDial benchmark by integrating 15 more datasets and introducing 38 novel atomic tasks. This results in the creation of 68 new downstream tasks by merging single-dimensional atomic tasks. Experimental results show the advantages of these compositional tasks.

**Questions For The Authors:**

See 3. and 4.

**Reasons To Accept:**

1. The paper introduces a unique framework that merges various 1-D atomic tasks, I think the motivation is clear and sound.

2. The established InstructDial benchmark is expanded with an addition of 68 novel downstream tasks.

**Reasons To Reject:**

1. While the paper introduces 68 new downstream tasks based on seven 2-D Cesar tasks, it mainly offers overall performance. This obscures the individual quality and contribution of each task. There's a possibility that only a few high-quality tasks drive the improvements, while the majority add little value.

2. The methodological clarity regarding CESAR's task combination into coherent instruction prompts remains ambiguous. It's essential to discern the reliability of these instructions and ensure task diversity, preventing overlapping or overly similar tasks.

3. The paper doesn't sufficiently differentiate itself from FLAN-T5, which already encompasses several dialogue tasks and also detailed task design. It would great if the paper could discuss the differences.

4. The generalization experiments of CESAR-FLAN-xxl, initialized from FLAN-xxl, are somewhat undermined. Given that FLAN-xxl already integrates around 400 datasets and over 1.5k tasks, and InstructDial also introduces several tasks for each dataset. The paper's claim of "unseen generalization" isn't wholly convincing unless both datasets and tasks from FLAN-xxl and InstructDial are explicitly excluded from evaluation. Besides, it may also leak data information for seen domains if different tasks come from the same datasets.

**Reproducibility:**

2: Would be hard pressed to reproduce the results. The contribution depends on data that are simply not available outside the author's institution or consortium; not enough details are provided.

**Reviewer Confidence:**

4: Quite sure. I tried to check the important points carefully. It's unlikely, though conceivable, that I missed something that should affect my ratings.

---

> ### Author Rebuttal · Authors · 2023-08-28
>
> We appreciate the detailed feedback from Reviewer "dek5". We are happy that they thought the **motivation of the paper was clear and sound**.
>
> CESAR centers on the generation of prompts and instructions within dialogues. Our principal innovation lies in its capacity to craft compositional task prompts, an aspect previously overlooked in prior studies in this domain.
>
> Below, we address each of the Reviewer's concerns:
>
> > While the paper introduces 68 new downstream tasks based on seven 2-D Cesar tasks, it mainly offers overall performance. This obscures the individual quality and contribution of each task. There's a possibility that only a few high-quality tasks drive the improvements, while the majority add little value.
>
> - **CESAR’s core contribution:** The central concept of the paper revolves around the incorporation of task composition rather than a mere increase in the number of tasks. Such compositions are not explored in any of the previous Instruction-tuning benchmarks, such as Flan, OPT-IML, etc., and particularly in dialog benchmarks, such as - InstructDial and DialogStudio.
>
> - **Data Mixing Strategy:** As our core proposal is in the automatic creation of compositional dialog instructions, we follow prior work in our data mixing strategy. The reviewer is correct in their understanding that certain datasets, and tasks could be overpowering the others. To alleviate this, previous work, such as [FLAN](https://arxiv.org/abs/2109.01652), [InstructDial](https://arxiv.org/abs/2205.12673), and [Super-NaturalInstructions](https://arxiv.org/abs/2204.07705), limit the overall quantity of samples per task. This restriction aims to ensure a balanced distribution of data. Moreover, in the [Flan-collection paper](https://arxiv.org/abs/2301.13688) authors highlight that while they performed ablations across prompt themes -- like CoT, templated, etc., they followed "practitioner’s intuition" and performed data balancing using max thresholds. We adopt the same strategy.
>
> - **Principled Data Mixing:** Identifying optimal task/data mixtures is a research area on its own and is not the primary focus of our work. For example [Chen et al.](https://arxiv.org/abs/2307.14430) show a systematic way to create a mixture in their recent paper.
>
> > The methodological clarity regarding CESAR's task combination into coherent instruction prompts remains ambiguous. It's essential to discern the reliability of these instructions and ensure task diversity, preventing overlapping or overly similar tasks.
>
> - **Reliability of Prompts:** The CESAR framework ensures reliable combined task instructions through its inherent structure. We emphasized this aspect throughout our work. Our main goal was to automate prompt generation for combined tasks. To achieve this, we developed a modular prompt design, which is shown in Figure 10-b in the appendix. This figure demonstrates a sample prompt created using the CESAR framework.
>
> - **Task/Prompt Diversity:** We agree with the reviewer that the paper lacks a specific analysis of task/prompt diversity. We value this feedback and intend to incorporate it in the paper's final version. Building on our previous efforts, we are eager to present an advanced diagram illustrating the task categories utilized by the CESAR framework, accompanied by the tasks showing the diversity of InstructDial++.
>
> > The paper doesn't sufficiently differentiate itself from FLAN-T5, which already encompasses several dialogue tasks and detailed task design. It would be great if the paper could discuss the differences.
>
> - The Flan-T5 introduces a variety of datasets and tasks in the flan collections, but only a handful pertains to conversational AI. Another such example is [OPT](https://arxiv.org/abs/2205.01068) which includes collections from multiple dialogue datasets, however, they are not publicly available.
>
> - Furthermore, Iyer et al. have pointed out that integrating dialog data negatively impacts the outcomes, leading them to exclude dialogue data when fine-tuning the final [OPT-IML model](https://arxiv.org/abs/2212.12017). This insinuates that dialog modality might benefit from a dedicated benchmark.
>
> - While some proponents have introduced public dialog benchmarks like InstructDial and DialogStudio, none of these datasets address compositions, which is the central focus of our paper.
>
> - All in all, we agree with the reviewer that these comparisons could be helpful. We will add a new table in the final version of the paper where we discuss differences between popular multi-tasking benchmarks --> Flan, OPT-IML, InstructDial, and ours.
>
> | Collection    | Model                       | Base        | Size       | Public | Prompt                  | # Tasks | # Data Points | Objective                                                         |
> |---------------|-----------------------------|-------------|------------|--------|-------------------------|---------|---------------|-------------------------------------------------------------------|
> | FLAN          | FLAN-T5<br>FLAN-PALM        | T5, PALM  | 10M to 540B | P   | Zero, Few, COT       | 1836    | 15M           | Covers General NLP Tasks                                          |
> | OPT-IML       | OPT-IML                     | OPT         | 30 to 175B  | P   | Zero,Few, COT       | 2067    | 18M           | Covers General NLP Tasks and Meta Learning Instructions           |
> | InstructDial  | DIAL-T0, DIAL-BART0       | T0, BART0 | 400M to 3B  | P   | Zero, Few               | 48      | 250k          | Covers Dialog Specific Tasks                                      |
> | CESAR         | CESAR                       | FLAN-T5     | 3B to 11B   | P   | Zero                      | 154     | 450k          | Covers Dialog Specific and Compositional Tasks                    |
>
>
>
> > The generalization experiments of CESAR-FLAN-XXL initialized from FLAN-XXL, are somewhat undermined. Given that FLAN-XXL already integrates around 400 datasets and over 1.5k tasks, InstructDial also introduces several tasks for each dataset. The paper's claim of "unseen generalization" isn't wholly convincing unless both datasets and tasks from FLAN-XXL and InstructDial are explicitly excluded from evaluation. Besides, it may also leak data information for seen domains if different tasks come from the same datasets.
>
> - Generalization on Compositions: Our focus on generalization pertains to the composition level rather than individual tasks. Thus, we are interested in assessing the model's ability to generalize to task compositions it hasn't been specifically trained on, irrespective of whether it was trained on the constituent tasks.
> - The reviewers' concern might be particularly applicable to the final column, where we assert that both tasks are novel (end-with generation and emotion-grounded generation). However, for Table 5, our finetuning doesn't encompass the entire InstrucdDial dataset; instead, it involves a subset of tasks.
> - Therefore, the primary consideration here should be whether FLAN includes a task resembling end-with generation or emotion-grounded response generation. While - - Flan models might have seen individual tasks and their corresponding prompts, the compositions are not present in Flan -- particularly for dialog tasks. This is indeed our main contribution.
> - We will edit the relevant parts of the paper again to clearly specify our generalization claims.

---

### Official Review · Reviewer_gUjh · 2023-08-11

**Soundness:** 4

**Excitement:**

4: Strong: This paper deepens the understanding of some phenomenon or lowers the barriers to an existing research direction.

**Paper Topic And Main Contributions:**

The paper introduces a new framework called CESAR for instruction-based multitasking in dialog applications. It addresses the challenge of handling complex instructions by automatically generating compositional tasks without manual intervention. The authors enhance the InstructDial benchmark with new datasets and tasks, utilizing CESAR to create compositional tasks with multiple constraints. The resulting benchmark, InstructDial++, comprises 86 basic tasks and 68 composite tasks across 63 datasets. Through experiments, the authors demonstrate that CESAR's compositional tasks enhance performance on both seen and unseen compositions, bridging the gap between publicly-available dialog models and proprietary AI assistants.

**Reasons To Accept:**

1. The proposed unified framework presents a novel idea.
2. The motivation and details of the approach have been described in great detail.
3. Abundant experiments have validated the authors' method.

**Reasons To Reject:**

I haven't identified any significant shortcomings of the paper.

**Reproducibility:**

3: Could reproduce the results with some difficulty. The settings of parameters are underspecified or subjectively determined; the training/evaluation data are not widely available.

**Reviewer Confidence:**

2: Willing to defend my evaluation, but it is fairly likely that I missed some details, didn't understand some central points, or can't be sure about the novelty of the work.

---

### Official Review · Reviewer_Lkwr · 2023-08-12

**Soundness:** 4

**Excitement:**

4: Strong: This paper deepens the understanding of some phenomenon or lowers the barriers to an existing research direction.

**Paper Topic And Main Contributions:**

The paper notes that closed-access large language models such as the ones used in chatGPT are much more capable than the open-source LLMs and it is reasoned that this gap can be bridged by training the open LLMs on large-scale complex demonstrations, specifically compositional instructions: instructions where output is grounded on a set of two or more constraints

The paper proposes a framework called CESAR through which the authors clearly define a generalized representation of providing instruction and dialogue components as inputs to a language model. They further argue that improving the availability of demonstration data invariably leads to better instruction tuning performance therefore they augment the existing InstructDial dataset with more datasets and tasks. The resulting dataset is called InstructDial++

The paper finetunes a FLAN model on this augmented dataset and it is shown that when compared to benchmark models used in InstructDial, as well as FLAN model trained on InstructDial the CESAR approach performs better.


**Questions For The Authors:**

A) The section starting from L184 is important to motivate the need for compositional tasks, however the details regarding this preliminary experimentation are not provided.

B) In L401 is it mentioned that CESAR selects and organizes 2-D tasks automatically, however it is not clear how this is implemented and how are infeasible tasks identified




**Reasons To Accept:**

The paper is generally well written and well structured. Although the introduction and description of CESAR were somewhat challenging to follow due to many new terms being introduced, the accompanying tables and figures helped in following along.

The authors have identified a critical gap between performance of closed and open-source LLMs in task-oriented dialogues and clearly delineated their motivations with preliminary experiments to support their hypothesis that compositional instructions can help improve performance on complex dialog tasks.

The CESAR framework is well defined and designed to be scalable across different tasks within task-oriented dialogue.  They have explained how n-D tasks are defined, how discriminative tasks can be incorporated and how they ensure order-invariance while linearizing their inputs

The InstructDial++ dataset is a meaningful contribution (the authors have committed to sharing the dataset on Github) and results from Table 3 - 4 indicate that fine-tuning FLAN model on InstructDial++ shows performance gains over a FLAN model trained on InstructDial

The quantitative results indicate that CESAR-FLAN performs better than existing approaches on a good number of atomic tasks and seems to out-perform other approaches in compositional tasks.


**Reasons To Reject:**

One of the primary motivations of the paper is to bridge the gap between closed LLMs and open LLMs, hence there should have been more comparison and evaluation between CESAR and GPT-3.5-turbo in the main paper. The authors have tabulated some results in the appendix section and they have carefully analyzed why the numbers might be deceiving, however it would be better to show several qualitative examples across CESAR and GPT-3.5-turbo and analyze trends in performance, it not clear whether the gap has been bridged or not

In L213 it is stated that scalable generation of compositional tasks is one aspect that CESAR aims to solve, however it is not clear in the paper how the compositional tasks are generated in a  scalable fashion using CESAR


**Reproducibility:**

4: Could mostly reproduce the results, but there may be some variation because of sample variance or minor variations in their interpretation of the protocol or method.

**Reviewer Confidence:**

4: Quite sure. I tried to check the important points carefully. It's unlikely, though conceivable, that I missed something that should affect my ratings.

**Typos Grammar Style And Presentation Improvements:**

Figure 3 is not clear, what metric does performance indicate and what does data size indicate, are they number of utterances or dialogues as a whole. The authors could provide more details on this preliminary experimentation that they performed.

---

> ### Author Rebuttal · Authors · 2023-08-28
>
> We appreciate the thorough feedback from Reviewer "Lkwr". It's encouraging to know that they recognized **the critical gap** our paper addresses, viewed **CESAR as a well-defined framework**, and saw the **InstructDial++ dataset as a meaningful contribution**.
>
> CESAR centers on the generation of prompts and instructions within dialogues. Our principal innovation lies in its capacity to craft compositional task prompts, an aspect previously overlooked in prior studies in this domain.
>
> Below, we address each of Reviewer's concerns:
>
> > One of the primary motivations of the paper is to bridge the gap between closed LLMs and open LLMs, hence there should have been more comparison and evaluation between CESAR and GPT-3.5-turbo in the main paper. The authors have tabulated some results in the appendix section and they have carefully analyzed why the numbers might be deceiving, however, it would be better to show several qualitative examples across CESAR and GPT-3.5-turbo and analyze trends in performance, it is not clear whether the gap has been bridged or not.
>
> - We agree with the reviewer that adding such qualitative examples could assist the reader in better comprehending the distinctions between CESAR and ChatGPT.
> - Following the reviewer’s suggestion In the final version we will add qualitative examples from GPT3.5, GPT4 and Claude models.
> - We will also enrich the quantitative results with more shots during prompting and also using GPT4 API. For a sample table with preliminary results please see the answer to Reviewer "8BXP".
> - We will also provide a fine-grained analysis of areas of vulnerability causing publicly available models (i.e. CESAR in the context of this work) to lag behind closed models (GPT models and Claude) in tasks involving composition. We add one such sample here to give a better idea of what to expect in the final version:
>
> **Prompt**:
>
> “Please generate a response to the following dialog context.
> The response should start with this initial phrase: “Absolutely . That’s”.
> The response should also contain the following keywords: “thing” and “flat”.
>
> Dialog Context:
>
> Speaker 1: What kind of place shall we rent ?
>
> Speaker 2: It should be close to the university . Neither of us are good at getting up in the mornings and closer it is , the later we can get up.
>
> Response: ”
>
> **CESAR**: Absolutely. That’s the most important thing and flat should
> be furnished.
>
> **GPT 3.5**: “Absolutely. That’s a valid point. Being near the university is definitely a crucial thing to consider, especially given our morning habits. The closer we are, the more likely it is that we can avoid rushing in the early hours. Another thing to think about is the type of place itself. Would a flat be suitable for us, or are you leaning more towards a house? Let's make sure we find a place that aligns with both our proximity needs and our preferences for the kind of living space.”
>
> **GPT 4**: Absolutely. That's the smartest thing to consider. A flat near the university would be perfect for us.
>
> **Claude v2**:  Absolutely. That's a flat thing to consider when looking for a place near the university. Being close by means we can sleep in a bit later and still make it to class on time.
>
> - Note that this prompt incorporates a compositional task that CESAR has been trained on whereas other models are prompted in zero-shot. In the final version of the paper, we shall also try unseen tasks that CESAR has not seen before.
>
> > In L213 it is stated that scalable generation of compositional tasks is one aspect that CESAR aims to solve, however, it is not clear in the paper how the compositional tasks are generated in a scalable fashion using CESAR
>
> > In L401 is it mentioned that CESAR selects and organizes 2-D tasks automatically, however, it is not clear how this is implemented and how are infeasible tasks identified
>
>
> - **Scalable Task Composition via Automation**: As explained in section 4.2, CESAR does not only unify dialog tasks in a certain prompt structure but it utilizes this structure in order to combine these dialog tasks as compositional tasks in an automatic manner.
> - This automation is ensured by defining specific rules (10 in the current implementation) that limit what kind of CESAR tasks are composable together. Our framework utilizes these rules in the Task composition Operation (Definition 2 in the paper) in order to create only feasible compositional tasks. We shall share the complete list of these rules in the final version of the paper. One sample rule is as follows which defines the rule to create ICAA-R compositions where the generation incorporates 2 actions (e.g. beginswith_controlled_generation+keyword_controlled_generation)
>     - ("ICA-R","ICA-R"):{"composition": "ICAA-R", "common_fields": ["dialog context","response"], "target_field":"resonse", "Instructions": [AR_INST]},
>     - “common fields” stands for modules in each task that is common and should be incorporated in the final compositional task.
>     - “target field” represents the output expected in both tasks so it should also be the output of the final compositional task.
>     - “final instructions” represent a list of instructions that may be used for the composition. These instructions are manually written for each compositional CESAR task only once (there 7 such tasks in total).
>     - These are simplified instructions (prompts) that direct the model to utilize the “common fields” -> This ensures that the automated combination is grammatical and there is no need for fine-grained human intervention.
>     - Since we share the same structure in all ICA-R tasks our framework ensures that this composition should work flawlessly for any type of composition that abides by the rule.
> - The scalability proposition stems from CESAR's inherent capacity for automation.
>
> **Infeasible Tasks in Composition**: Without this automation, incorporating new tasks would entail manual identification by human annotators of potential compositional links with prior tasks, followed by the creation of new prompts, again carried out by human annotators. The manual identification would be necessary because composing every task together would result in an infeasible task (see Figure 11 for such an example).
> - Moreover even generating only feasible tasks with the assumption of an oracle, this naive composition results in worse results when compared to the CESAR framework (see Table 5 for results).
>
> **How does CESAR avoid Infeasible Tasks?**: In the section before, it was highlighted that CESAR sidesteps tasks that aren't feasible by leveraging 10 specific manually written rules.
> - Essentially, these rules specify which 1-D CESAR tasks can be combined. The primary advantage of CESAR lies in its ability to use these high-level rules to combine 86 distinct 1-D tasks, yielding 68 viable 2-D tasks. In the absence of this structured approach and rules, human annotators would be burdened with conducting 86x86 manual evaluations to ascertain the compositionality of these two tasks.
> - We acknowledge that in its current form, it is hard to understand from the paper how infeasible tasks are avoided by CESAR framework. We shall edit the relevant parts of the paper to ease understanding in the final version.
>
> > The section starting from L184 is important to motivate the need for compositional tasks, however, the details regarding this preliminary experimentation are not provided.
>
> - We understand the reviewers' concern that there are not enough experimentation details in this section. Our original intention was to only provide a brief section highlighting our motivation for the need for compositional prompts and delve into the actual experiments (with details) in Section 6. However, we agree that the section details can be made more enriched. We shall, thus, revise the section to make the experimental setup more explicit.
>
> - We shall add the details in the final version of the paper. This shall include but should not be limited to sample manual prompts used for compositions, and the amount of data used for training and testing.

---

### Meta-Review · Area_Chair_MKEK · 2023-09-19

**Recommendation:** 4

**Metareview:**

Based on the provided reviews, the paper has several strengths. It presents a novel framework called CESAR that addresses the challenge of handling complex instructions in dialog applications. The motivation and details of the approach are described in great detail, and the paper provides abundant experiments that validate the authors' method. The enhanced InstructDial benchmark, InstructDial++, is considered a meaningful contribution. The results show that CESAR-FLAN performs better than existing approaches on both atomic and compositional tasks.

However, there are also some concerns raised by the reviewers. One reviewer questions the lack of comparison and evaluation between CESAR and GPT-3.5-turbo, and suggests providing qualitative examples to analyze the performance gap. Another reviewer suggests providing more transparency in the generation of compositional tasks and addressing the ambiguity in task combination within CESAR. It is also suggested to differentiate the paper from FLAN-T5 and provide more thorough evaluation and comparisons with existing methods or baselines. Furthermore, one reviewer challenges the assumption that open-source models are less capable of handling compositional tasks and raises concerns about the dataset used for evaluations.

Overall, the paper is well-written and well-structured, and provides sufficient support for its main claims. It deepens the understanding of instruction-based multitasking in dialog applications and lowers the barriers to this research direction. However, there are some areas that need improvement, such as providing more detailed comparisons, addressing concerns about task generation and combining, and clarifying the assumptions and dataset used for evaluation.

---

### Decision · Program_Chairs · 2023-10-07

**Decision:**

Accept-Main

**Comment:**

Based on the provided reviews, the paper has several strengths. It presents a novel framework called CESAR that addresses the challenge of handling complex instructions in dialog applications. The motivation and details of the approach are described in great detail, and the paper provides abundant experiments that validate the authors' method. The enhanced InstructDial benchmark, InstructDial++, is considered a meaningful contribution. The results show that CESAR-FLAN performs better than existing approaches on both atomic and compositional tasks.

However, there are also some concerns raised by the reviewers. One reviewer questions the lack of comparison and evaluation between CESAR and GPT-3.5-turbo, and suggests providing qualitative examples to analyze the performance gap. Another reviewer suggests providing more transparency in the generation of compositional tasks and addressing the ambiguity in task combination within CESAR. It is also suggested to differentiate the paper from FLAN-T5 and provide more thorough evaluation and comparisons with existing methods or baselines. Furthermore, one reviewer challenges the assumption that open-source models are less capable of handling compositional tasks and raises concerns about the dataset used for evaluations.

Overall, the paper is well-written and well-structured, and provides sufficient support for its main claims. It deepens the understanding of instruction-based multitasking in dialog applications and lowers the barriers to this research direction. However, there are some areas that need improvement, such as providing more detailed comparisons, addressing concerns about task generation and combining, and clarifying the assumptions and dataset used for evaluation.